# Representative Ranking for Deliberation in the Public Sphere

**Manon Revel** [* 1 2]   **Smitha Milli** [* 1]   **Tyler Lu** [3]   **Jamelle Watson-Daniels** [1]   **Max Nickel** [1]

## Abstract

Online comment sections, such as those on news sites or social media, have the potential to foster informal public deliberation, However, this potential is often undermined by the frequency of toxic or low-quality exchanges that occur in these settings. To combat this, platforms increasingly leverage algorithmic ranking to facilitate higher-quality discussions, e.g., by using civility classifiers or forms of prosocial ranking. Yet, these interventions may also inadvertently reduce the visibility of legitimate viewpoints, undermining another key aspect of deliberation: representation of diverse views. We seek to remedy this problem by introducing guarantees of representation into these methods. In particular, we adopt the notion of *justified representation* (JR) from the social choice literature and incorporate a JR constraint into the comment ranking setting. We find that enforcing JR leads to greater inclusion of diverse viewpoints while still being compatible with optimizing for user engagement or other measures of conversational quality.

## 1. Introduction

In theories of deliberative democracy (Habermas, 1991; Cohen, 2005; Fishkin, 2009; Bächtiger et al., 2018), thoughtful and constructive public discourse is key for a well-functioning polity. Online platforms, such as comment sections on news sites or social media, present an opportunity to expand this kind of public sphere deliberation (Helberger, 2019; Landemore, 2024). These digital spaces enable interactions between individuals who might never meet in person, potentially exposing them to a broader array of viewpoints. However, in reality, these online discussions often deteriorate into low-quality or toxic exchanges (Nelson et al., 2021; Kim et al., 2021). In this work, we aim

to bring online conversations closer towards the ideals of deliberation, leveraging algorithmic tools in doing so.

What does it take for a discussion to count as *deliberative*? Deliberative democracy ideals generally demand at minimum (i) certain aspects of conversational quality (e.g., respect, reasoned arguments, etc) *and* (ii) the representation of diverse voices. For example, among other ideals, Fishkin & Luskin (2005) claim that deliberative discussion must be (i) *conscientious*, i.e., "the participants should be willing to talk and listen with civility and respect," and (ii) *comprehensive*, i.e., "all points of view held by significant portions of the population should receive attention." Similarly, in their review of the 'second generation of deliberative ideals,' Bächtiger et al. (2018) highlight the ongoing importance of (i) respect, and (ii) the evolving view of *equality* as the equality of opportunity to political influence (Knight & Johnson, 1997), which requires that people with diverse viewpoints be able to receive attention to their perspectives.

Algorithmic moderation and ranking have become key tools for facilitating conversational quality in online discussions (Kolhatkar et al., 2020). For example, Google Jigsaw's Perspective API comment classifiers have been used by prominent news organizations, such as the New York Times and Wall Street Journal, to filter or rerank comments to improve civility (Lees et al., 2022; Saltz et al., 2024). More recently, a line of work on *bridging systems* advocating ranking aims at *bridging* different perspectives and reducing polarization (Ovadya & Thorburn, 2023). Bridging-based ranking has been used to in several high-impact applications such as ranking crowd-sourced fact checks on X, Instagram, Threads, Facebook, and TikTok (Wojcik et al., 2022; Meta, 2025; TikTok, 2025) and selecting comments in collective response systems like Pol.is and Remesh (Small et al., 2021a; Konya et al., 2023a; Huang et al., 2024).

While these interventions may improve conversational quality, the goal of ensuring representation in online discussions is less studied. In fact, solely focusing on conversational quality might inadvertently result in the censorship of certain groups, in the sense that their comments may be filtered out or do not appear high enough in the comment ranking to gain visibility. For example, civility classifiers have been shown to disproportionately flag comments written in African-American English (Sap et al., 2019) and comments

---

*Equal contribution   [1]Meta FAIR, New York, NY, USA [2]Berkman Klein Center, Harvard University, Cambridge, MA, USA [3]Meta, New York, NY, USA. Correspondence to: Manon Revel <manon@meta.com>, Smitha Milli <smilli@meta.com>.

*Proceedings of the 42nd International Conference on Machine Learning*, Vancouver, Canada. PMLR 267, 2025. Copyright 2025 by the author(s).

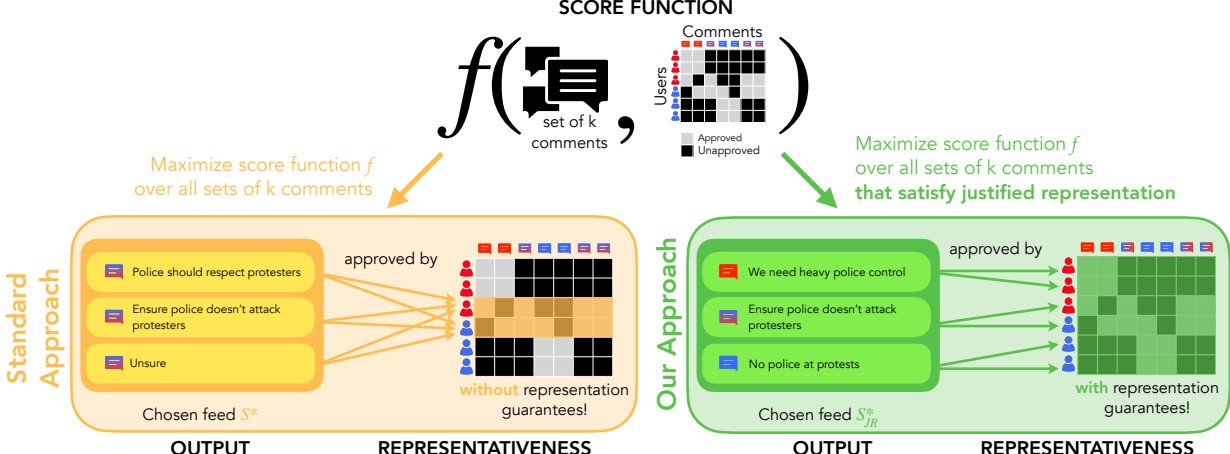

*Fig. 1.* **Our approach compared to the standard approach to ranking comments.** In this example, a platform with $n = 6$ users wants to select $k = 3$ comments to highlight while optimizing a given score function $f$ (e.g. engagement, civility, diverse approval, etc). This example is inspired by our experiments ranking comments related to campus protests in Sec 6. In the standard approach, the platform simply selects the $k$ comments with the highest score (in this case, diverse approval across the red and blue users). However, this leads to only two users having a comment that they approve of in the selected set $S^*$. The two remaining red and blue users do not receive representation despite the fact that they each form a group large enough to warrant representation by proportionality (each of size $\lceil n/k \rceil \geq \lceil 6/3 \rceil = 2$) and are minimally cohesive (have at least one item that they agree upon in common). On the other hand, in our approach, the platform picks the $k$ comments that maximize the score while satisfying *justified representation*, and guarantees that all cohesive groups of size at least $n/k$ are represented in the resulting set $S^*_{JR}$.

discussing encounters with racism (Lee et al., 2024). In the context of diverse approval, the groups being bridged have a large effect on what content gets shown. Commonly, these groups correspond to political groups like left- or right-leaning users (Wojcik et al., 2022). Therefore, it is plausible that diverse approval could give more visibility to moderate comments, while failing to provide representation of others who hold more ideologically diverse views.[1]

In this paper, our goal is to broaden the scope of deliberative ideals examined in the algorithmic facilitation of online discussions. In particular, we extend the existing focus on conversational quality to also include ideals of representation. In brief, our contributions are the following:

1. **Theoretical framework.** We contribute a theoretical framework in which the platform selects the top-ranked comments by maximizing a score function $f$ (e.g. civility, bridging objectives) subject to a representation constraint known as *justified representation* (JR) (Aziz et al., 2017) from the social choice literature on approval voting in multi-winner elections.

2. **Theoretical analysis.** We theoretically analyze how

the JR constraint affects the platform's ability to maximize different classes of score functions. Despite negative results in worst-case settings, we show that in the natural setting where users' opinions cluster into a few groups, enforcing representation does not come at a price to other objectives such as civility or user engagement (§ 5).

3. **Method and empirical analysis.** Leveraging approximation algorithms from social choice, we implement a JR constraint for ranking comments on Remesh[2] related to campus protests. We find that enforcing JR significantly enhances representation without compromising other measures of conversational quality. Specifically, we evaluate the price of JR for metrics beyond simple engagement, including diverse approval and content-based bridging classifiers (§ 6), and find a low price in all settings, as predicted by our theoretical results (§ 5).

## 2. Related work

Unlike aggregative models of electoral democracy (Manin, 1997), which rely on citizens' periodic votes to express political consent (Landemore, 2020), deliberative democracy

---

[1]Or, imagine an online platform popular with fans of the Celtics and Knicks (two U.S. basketball teams) where the only posts that get diverse approval are those about Kadeem Allen (a former player in both franchises). Bridging-based ranking à la Ovadya & Thorburn (2023) may overly represent the Kadeem Allen posts, downgrading legitimate interest of the broader fan-bases.

[2]Remesh (Konya et al., 2023a) is a popular collective response system (Ovadya, 2023) used frequently by governments, non-profits, and corporations to elicit opinions from a collective at scale.

anchors political legitimacy in discursive processes through which citizens continuously participate and consent to the polity's activity (Habermas, 2015). Deliberative democracy scholars have identified certain ideals that democratic deliberation should strive to uphold (e.g. respect, absence of coercive power, etc), and Bächtiger et al. (2018) review how conceptualizations of these ideals have evolved over the years. More recently, scholars have begun exploring the potential for online spaces to democratize and scale deliberative discourse (Buchstein, 1997; Helberger, 2019; Gelauff et al., 2023).

Algorithmic interventions for comment ranking focus primarily on promoting conversational quality (Kolhatkar & Taboada, 2017; Saltz et al., 2024; Lees et al., 2022; Ovadya & Thorburn, 2023; Jia et al., 2024; Piccardi et al., 2024), but without guarantees on representation of viewpoints—another key ideal of deliberation. To incorporate such guarantees into these methods, we focus on selecting the *set* of top $k$ comments, which allows us to leverage notions of proportional representation. While other research on diversity in recommender systems also focuses on selecting optimal *sets* of items, diversity in this context is typically defined based on the content of items, e.g., choosing videos from a variety of topics (Kunaver & Požrl, 2017; Zhao et al., 2025). In contrast, our goal here is to select a set of comments that ensures *representation*, thereby focusing on the perspective of users rather than the content diversity of the items.

To formalize representation, we borrow the social-choice-theoretic concept of *justified representation* (JR), developed by Aziz et al. (2017) in the context of approval voting in multi-winner elections. Justified representation is an axiom that formalizes a notion of proportionality: it guarantees that every large enough group of users that has shared preferences (approves of at least one comment in common) is allocated at least one of the top comments. While prior work has focused on selecting or generating comments subject to JR or extensions of JR (Halpern et al., 2023; Fish et al., 2024; Bernreiter et al., 2024), these works do not consider the optimization of other exogenous objectives (e.g. content-based classifiers for civility or other attributes) that are a key component of real-world comment ranking. Therefore, we study a constrained optimization problem where the platform selects the top comments based on optimizing a score function (e.g. civility, diverse approval, engagement, etc) subject to JR.

For the comment ranking setting, in particular, JR affords several advantages over notions of demographic or social representation more commonly used in the algorithmic fairness literature (Barocas et al., 2023; Chasalow & Levy, 2021). We discuss these advantages further in Appendix A, but here, we highlight two benefits. First, unlike most algorithmic fairness methods, JR does not require inference

of any demographic or social labels, making it compatible with the privacy and legal constraints of real-world platforms (Holstein et al., 2019; Veale & Binns, 2017). Second, JR automatically adapts to the groups that are relevant in different contexts. For instance, the groups relevant to represent in the discussions about the NYC budget planning process differ from those in a post about a basketball game between the Celtics and the Knicks. On real-world platforms that have a diverse range of content, the fact that JR naturally focuses on the relevant groups is a significant advantage.

The primary theoretical question we analyze is: what is the price of enforcing JR, with respect to the ability to optimize other score functions? In the recommender system context, it is essential that a representation constraint be compatible with other objectives of interest for the platform (e.g. civility of comments). The need to optimize other objectives well is also why we start with the fundamental JR axiom rather than its stronger extensions (Aziz et al., 2017; Sánchez-Fernández et al., 2017; Skowron et al., 2017; Brill & Peters, 2023; Fish et al., 2024; Peters & Skowron, 2020). Nevertheless, in our empirical experiments, we find that all but one of our JR committees also satisfies the stronger EJR+ axiom (Brill & Peters, 2023), consistent with other studies that find minimal differences between these axioms in real-world settings (Boehmer et al., 2024).[3]

The closest related theoretical work is by Elkind et al. (2022; 2024) who investigate the price of JR with respect to utilitarian social welfare (the number of selected comments that each user approves of, summed over users). Other related work includes Skowron et al. (2017) who study the performance of various approval-based committee rules with respect to social welfare and coverage and Bredereck et al. (2019) who establish that maximizing social welfare or coverage with respect to JR is NP-hard. In contrast to Elkind et al. (2022), we analyze the price of JR for more general classes of score functions, which are relevant in the context of recommender systems where objectives beyond user approval are also important, as well as constraints on the user approval matrix. We show that in the common setting when users' preferences cluster into a few groups, the price of JR, even with respect to arbitrary objective functions, is typically negligible (§ 5). These findings also relate to Faliszewski et al. (2018) who use clustering as an empirical heuristic to achieve fully proportional representation.

## 3. Problem Setting

In this section, we define the comment ranking problem, the axiom of justified representation and the price function used to examine how ensuring representation might affect

---

[3] See also Bardal et al. (2025) for similar findings beyond the approval voting setting.

the ability to maximize other objectives.

## 3.1. Model

For $m \in \mathbb{N}$, we write $[m] = \{1, \ldots, m\}$. Let $[m]$ be the set of all comments and $[n]$ be a set of users. The platform must select a subset $S \subseteq [m]$ of $k$ comments to highlight.[4] Each user $u \in [n]$ approves a set $A_u \subseteq [m]$ of comments and $\mathcal{A}_n = \{A_1, \ldots, A_n\}$ is the profile of approved comments across all users. In practice, a user's approval set might be defined as the set of comments that they upvote, like or react to. In all, each problem is characterized by a tuple $\mathcal{I}_{m, \mathcal{A}_n, k} = \langle m, \mathcal{A}_n, k \rangle$ of $n$ approval sets over $m$ comments, from which $k$ comments are selected.

## 3.2. Scoring Rule

Comments are ranked based on a scoring rule: each comment $i \in [m]$ is assigned a score $f(i, \mathcal{A}_n) \geq 0$. The score of a set $S \subseteq [m]$ is additive; $f(S, \mathcal{A}_n) = \sum_{i \in S} f(i, \mathcal{A}_n)$. For notational simplicity, we occasionally write $f(i)$ or $f(S)$ and drop the potential dependence on the approval profile $\mathcal{A}_n$. We use $S^*(\mathcal{I}_{m, \mathcal{A}_n, k})$ to denote the[5] subset of $[m]$ with highest score for the instance $\mathcal{I}_{m, \mathcal{A}_n, k}$ :

$$f(S^*(\mathcal{I}_{m, \mathcal{A}_n, k})) = \max_{S \subseteq [m], |S| = k} f(S). \tag{1}$$

### 3.2.1. GENERAL SCORING RULES

In general, we consider arbitrary scoring rules $f$ that are potentially independent of the approval profile $\mathcal{A}_n$. For example, a scoring rule $f_C$ could be a classifier that assigns a score to a comment based solely on its textual content. A notable example are the Perspective API classifiers, a suite of models that have been used by prominent news organizations like the New York Times and Wall Street Journal, for algorithmic filtering and ranking of comments. The Perspective API models assess comments for attributes such as toxicity, compassion, reasoning, and more, relying solely on the comment text for classification.

### 3.2.2. APPROVAL DEPENDENT SCORING RULES

We also consider a natural class of scoring rules that are *approval dependent* in that the score of comment $i$ cannot decrease if a new user approves of it, holding all else equal. Formally, we write $A'_u \succeq_{(i)} A_u$ if $A'_u = A_u \cup \{i\}$. Then, we define an approval dependent scoring rule as the following.

**Definition 1** (Approval Dependent)**.** A scoring rule $f(i, \mathcal{A}_n)$

---

[4]Work on diversity in recommender systems also focuses on sets of items, but with the goal of showing "diverse" items where diversity is typically defined based on the content of the items, e.g., showing videos spanning different topics or genres (Kunaver & Požrl, 2017; Zhao et al., 2025). In contrast, our focus is on selecting sets of items (comments) to ensure that the top items provide a degree of proportional representation to users.

[5]If there are multiple highest-score sets, we pick one randomly. Our results do not depend on the tie-breaking rule for $S^*$.

is approval dependent if, for all items $i$ not in any approval set of $\mathcal{A}_n$ (that is $i \notin \bigcup_{u=1}^{n} A_u$), then we have $f(i, \mathcal{A}_n) = 0$.

*Engagement.* A simple example of an approval dependent scoring function is the utilitarian scoring rule in which each item's score equals the total number of users that approve it:

$$f_{\text{eng}}(i, \mathcal{A}_n) = |\{u : i \in A_u\}|. \tag{2}$$

In prior work rooted in the multi-winner election setting, the utilitarian scoring rule has been referred to as 'social welfare' (Elkind et al., 2022). In the context of comment ranking, the approval profile $\mathcal{A}_n$ would, in practice, likely be defined by user engagement. For example, a user might be said to 'approve' of a comment if they upvote or like it. Therefore, in this setting, it may be more accurate to think of the utilitarian scoring rule as an engagement-maximizing scoring rule that social media platforms are incentivized to optimize for. As such, we refer to this scoring rule as the engagement scoring rule.

*Diverse approval* is another example of an approval dependent scoring rule, where the score of a comment reflects the level of approval it receives across diverse groups of users. These groups could be pre-defined, e.g., based on demographic characteristics, or more commonly, learned from the data. Diverse approval is motivated by research finding that, compared to pure engagement, diverse approval tends to correlate with comments that are less toxic, more informative, and higher quality (Ovadya & Thorburn, 2023; Wojcik et al., 2022). Diverse approval has been used to select and rank user-generated content in high-impact applications such as ranking crowd-sourced fact checks on social media platforms (Wojcik et al., 2022; TikTok, 2025; Meta, 2025), and selecting comments in collective response systems like Pol.is and Remesh (Small et al., 2021a; Konya et al., 2023a; Huang et al., 2024).

Typically, in a diverse approval metric, users are partitioned into $\gamma$ non-overlapping groups $G_1, \ldots, G_\gamma \subseteq [n]$. The number of groups is quite small in practice (Wojcik et al., 2022; Huang et al., 2024). One simple diverse approval metric, which we refer to as *maximin diverse approval* (MDA), scores each comment by its minimum approval rate across groups:

$$f_{\text{DA}}(i, \mathcal{A}_n) = \min_{g \in [\gamma]} \frac{1}{|G_g|} |\{u : u \in G_g, i \in A_u\}|. \tag{3}$$

Other variants have also been used such as the product of the approval rate across groups (Small et al., 2021a; Huang et al., 2024) or a softmax version of minimax diverse approval (Konya et al., 2023a). Note that these examples satisfy a property that is strictly stronger than approval dependency, that of approval monotonicity where an approval monotonic function is an approval dependent function such that for all approval profiles $A'_n$, and $\mathcal{A}_n$ with

$A'_u \succeq_{(i)} A_u$ for some $u$ and $A'_v = A_v$ for all $v \in [n] \setminus \{u\}$, then $f(i, \mathcal{A}'_n) \geq f(i, \mathcal{A}_n)$.

### 3.3. Justified Representation

The problem of selecting $k$ items from a set $[m]$ based on individual's approval ballots is well studied in the context of multi-winner social choice, where the $k$ "items" are candidates to be selected for a committee.[6] In order to ensure fairness, in the sense that all large groups with cohesive preferences receive some amount of representation, (Aziz et al., 2017) axiomatically formalized the idea of *justified representation*. Following (Elkind et al., 2022)'s exposition, and (Bredereck et al., 2019)'s notion of justifying sets, assume we are given an instance $\mathcal{I}_{m,\mathcal{A}_n,k}$ of $n$ approval sets over $m$ comments, from which the platform must select $k$ comments.

**Definition 2** (Cohesiveness). A group of users $G \subseteq [n]$ is said to be *cohesive* if all the users in $G$ approve at least one common item: $\cap_{u \in G} A_u \neq \emptyset$.

**Definition 3** (Representativeness). A set of items $S \subseteq [m]$ is further said to *represent* a group $G$ of users if at least one user from $G$ approves of at least one item in $S : \exists u \in G$ such that $A_u \cap S \neq \emptyset$.

**Definition 4** ($n/k-$justifying set). A set of items $S \subseteq [m]$ is a $n/k-$justifying set if it represents every cohesive group of at least $n/k$ users.

The concept of Justified Representation (JR) ensures that large enough cohesive groups are minimally represented.

**Definition 5** (Justified Representation). An $n/k$-justifying set $S \in [m]$ is said to satisfy justified representation over $\mathcal{I}_{m,\mathcal{A}_n,k}$ if $|S| = k$.

In the comment ranking setting, a cohesive group is a set of users that approves of at least one comment in common. If a cohesive group contains least $n/k$ users, then by the principle of proportionality, since we are selecting $k$ comments to highlight, this group is considered deserving representation. A cohesive group $G \subset [n]$ is said to be represented by a set of comments $S \subset [m]$ if at least one user in the group $G$ approves of a comment in $S$.

---

[6]An intuitive approach would be to select the $k$ items with the largest approval score $|\{u \in [n] \mid i \in A_u\}|$. However, this method tends to favor items supported by the majority, potentially excluding minority groups from representation. Let a world with 60 people who approve the 10 items in set $A$ and another 40 people who approve of a distinct set $A'$ of 10 items. If a committee of size 10 is selected based on approval scores, the winning committee, $A$ would fail to represent $40\%$ of the world. Instead, a committee composed of 6 items from $A$ and 4 items from $A'$ would respect an intuitive notion of proportionality.

### 3.4. The Price of Justified Representation

Justified representation is not guaranteed to be compatible with the objectives of general scoring rules.[7] We formally analyze the question: what price must be paid in the total score if we require that selected items satisfy JR? We define the optimal JR set $S^*_{JR}(\mathcal{I}_{m,\mathcal{A}_n,k}, f)$ as the set that maximizes the score among all sets that satisfy justified representation[8]:

$$S^*_{JR}(\mathcal{I}_{m,\mathcal{A}_n,k}, f) = \underset{S \subseteq [m], |S|=k}{\arg\max} \; f(S) \tag{4}$$
$$\text{s.t. } S \text{ satisfies JR.}$$

Next, we define the price of JR as the ratio between the maximum (unconstrained) score and the maximum JR score, both for a specific instance, and as the maximum over all instances for a given $k$.

**Definition 6** (The Price of Justified Representation). We define the price of justified representation on an instance (Eq. 5) and over all instances with set of size $k$ (Eq. 6) as

$$P(\mathcal{I}_{m,\mathcal{A}_n,k}, f) = \frac{f(S^*(\mathcal{I}_{m,\mathcal{A}_n,k}, f))}{f(S^*_{JR}(\mathcal{I}_{m,\mathcal{A}_n,k}, f))}, \tag{5}$$
$$P(k, f) = \max_{\mathcal{I}_{m,\mathcal{A}_n,k}} P(\mathcal{I}_{m,\mathcal{A}_n,k}, f). \tag{6}$$

This formulation follows that of Elkind et al. (2022) who analyze the price of JR $P(k, f_{\text{eng}})$ for specifically the engagement scoring function $f_{\text{eng}}$. In later sections, we will also analyze a probabilistic version of the price of JR in Sec. 5.2 and prove high probability bounds over the distribution of instances in addition to our worst-case analysis of the price of JR.

Finally, when either the instance and scoring function is clear from context, we short-hand $S^*(\mathcal{I}_{m,\mathcal{A}_n,k}, f)$ and $S^*_{JR}(\mathcal{I}_{m,\mathcal{A}_n,k}, f)$ by $S^*$ and $S^*_{JR}$, respectively. Similarly, when the scoring function $f$ is clear from context, we short-hand $P(\mathcal{I}_{m,\mathcal{A}_n,k}, f)$ and $P(k, f)$ as $P(\mathcal{I}_{m,\mathcal{A}_n,k})$ and $P(k)$, respectively.

## 4. The General Price of JR

We first study the price of JR without imposing any constraints on the approval profiles of users. We analyze the price of JR for general scoring functions and for approval-dependent scoring functions. Unfortunately, we find that for either class of functions, the price of JR can be quite high. All proofs are deferred to Appendix C.

First, we establish that for general scoring functions, which includes functions that are independent of the approval profile such as content classifiers, the price of JR need not be bounded.

---

[7]We show how JR conflicts with diverse approval in Fig. B.1.
[8]Our results hold irrespective of random tie-breaking.

**Proposition 1.** *There exists a function $f$ such that $P(k, f)$ is unbounded.*

The result is not so surprising as, in general, if the scoring function can be independent of the approval profile (as with a classifier $f_C$), we need not expect that it gives representation to cohesive groups which are defined by their common approval. This exemplifies the potential negative externalities of using content classifiers that do not account for the users' own approval of different comments.

We next turn to analyzing the price of JR for approval dependent rules (Def. 1) which require that if an item has a positive score that someone must have approved of it. We show that the price of JR is $k$ for such rules. In fact, the result is true even if we restrict ourselves to bridging scoring rules used in practice like maximin diverse approval.

**Theorem 1.** *Assume $\gamma > 1$ and there exists an item with positive score. For any scoring function $f$ that is approval-dependent, $P(k, f)$ is at most $k$ and is equal to $k$ for certain approval-dependent scoring rules such as maximin diverse approval $f_{DA}$.*

For the comment ranking setting, a price of $k$ may still be high. For example, suppose we use maximin diverse approval to score comments and select 10 comments to highlight. Our results imply that, in the worst-case, enforcing representation can mean that the optimal JR set $S^*_{JR}$ attains $1/10$-th the diverse approval that the optimal unconstrained set $S^*$ does—an undesirable outcome if diverse approval is a priority. In the next section, we show that adding a natural assumption on the clustering of user preferences yields much lower bounds on the price of JR for all scoring rules.

## 5. The Price of JR in a Clustered Setting

Given the potential high price of JR when users' approval profiles can be arbitrary, we now turn to asking whether there are natural restrictions where the price of JR is low. We analyze one such setting—the setting in which users' preferences cluster into a few groups. This scenario is not only common in practice but also is the setting where bridging interventions, which aim to bridge over a few polarized groups, are most relevant. In particular, we assume that users can be partitioned into $\gamma$ groups $G_1, \ldots, G_\gamma \subseteq [m]$ on the basis of their approval profiles. We refer to these groups as *divided groups* to contrast them from the cohesive groups that are relevant for justified representation.

We show that when $\gamma$ is small and the groups display high within-group homogeneity, the price of JR is low. We operationalize within-group homogeneity in two distinct ways, showing that the price of JR remains consistently low under both interpretations. These results are notable, because, in practice, when bridging interventions are implemented, (a) $\gamma$ *is* small and (b) divided groups are often determined

by clustering or matrix factorization of the user approval matrix, resulting in groups that are relatively distinct and homogeneous (Wojcik et al., 2022; Small et al., 2021b).

### 5.1. Homogeneity as Cohesiveness
We first illustrate the price of JR in the setting where within-group homogeneity is operationalized as cohesiveness in the JR sense. That is, users are partitioned into $\gamma$ non-overlapping groups $G_1, \ldots, G_\gamma \subset [m]$ and each of these groups is cohesive.

**Theorem 2.** *Assume that $k > 1$ and users are partitioned into $\gamma$ cohesive groups $G_1, \ldots, G_\gamma$ where $1 \leq \gamma < k$. For any scoring rule $f$, the price of justified representation $P(k, f)$ is at most $\frac{k}{k-\gamma}$, and is exactly that for certain scoring rules such as maximin diverse approval $f_{DA}$.*

Assuming there are only a few divided groups, i.e. $\gamma$ is small, the price of JR of $k/(k - \gamma)$ is close to 1 (the lowest possible value).

### 5.2. Homogeneity as Low Dispersion
Operationalizing homogeneity as cohesiveness was illustrative, but in real-world scenarios, it may be unlikely for all the members in a divided group to approve on one comment in common. For a more realistic model, we now operationalize within-group homogeneity in the context of a common statistical model of preferences. In particular, we assume that users' preferences over items are drawn from a Mallows mixture model (Awasthi et al., 2014; Liu & Moitra, 2018), where each divided group has its own component in the mixture. We then operationalize homogeneity by the level of dispersion $\phi \in [0, 1]$ of individual preferences from their group-specific central ranking. In this setting, we find that with high probability, the price of JR is at most $O\left(k/\left(k - \gamma \lceil (\log k)/\log(1/\phi) \rceil\right)\right)$, which approaches our previous bound of $O(k/(k - \gamma))$ as within-group homogeneity increases, i.e., as $\phi \to 0$. Let us begin by defining the Mallows model.

**Definition 7** (Mallows model)**.** The Mallows model $M(\phi, \sigma)$ is described by a central reference ranking $\sigma$ and a dispersion parameter $\phi \in [0, 1]$. The probability of observing a preference ranking $\pi \sim M(\phi, \sigma)$ is given by

$$\Pr(\pi; \phi, \sigma) = \phi^{d(\pi, \sigma)}/Z(\phi), \qquad (7)$$

where $d$ is the Kendall-tau distance, which measures the number of pairs of items $(x, y)$ that two rankings disagree on (i.e. $x$ preferred to $y$ in one ranking and $y$ to $x$ in the other ranking). Finally $Z(\phi) = 1 \cdot (1 + \phi) \cdot (1 + \phi + \phi^2) \cdots (1 + \cdots + \phi^{m-1})$ is the normalization constant.

Note that when the dispersion parameter $\phi$ is equal to zero, then the model generates the reference ranking with probability one. At the other extreme, when the dispersion parameter $\phi$ is equal to one, then the model becomes a uniform

distribution over rankings. Thus, low values of $\phi$ indicate greater homogeneity.

We model preferences as being drawn from a Mallows mixture model where each divided group has its own component.

**Definition 8** (Mallows mixture model). The Mallows mixture model $\mathcal{M}_\gamma(\phi, \sigma, \lambda)$ combines $\gamma$ Mallows models. Let $\phi = (\phi_1, \ldots, \phi_\gamma)$ be a collection of dispersion parameters, $\sigma = (\sigma_1, \ldots, \sigma_\gamma)$ be a collection of reference rankings, and $\lambda = (\lambda_1, \ldots, \lambda_\gamma)$ be the mixing proportions where for all $i \leq \gamma$, we have $\lambda_i \geq 0$ and $\lambda_1 + \cdots + \lambda_\gamma = 1$. The mixture model is defined by the probability mass function

$$\Pr(\pi; \phi, \sigma, \lambda) = \sum_{i=1}^\gamma \lambda_i \cdot \Pr(\pi; \phi_i, \sigma_i). \qquad (8)$$

A sample from the Mallows model represents a user $u$'s ranked preference $\pi_u$ over all items. For the approval voting setting, we assume that user approves of the top $\tau$ items in their ranking $\pi$, i.e., $A_u = \pi_u^{-1}([\tau])$. We can obtain the following analytical upper bound on the price of JR, which is independent of any particular algorithmic strategy.

**Theorem 3.** *Suppose that each user $u$'s ranking $\pi_u$ is drawn from the Mallows mixture model $\mathcal{M}_\gamma(\phi, \sigma, \lambda)$ and their approval set $A_u$ is equal to the top $\tau$ items in their ranking, denoted by $\pi_u^{-1}([\tau])$. Let $\phi_{\max} = \max_i \phi_i$ be the maximum dispersion parameter for any group (component of the mixture). Then, for any scoring function $f$, and for all $\phi_{\max} \in [0,1]$, with probability $1 - \delta$ over the i.i.d. draws of user rankings $\pi_u$, $u \in [n]$, that maps to the approval profile $\mathcal{A}_n$, we have*

$$P(\mathcal{I}_{m, \mathcal{A}_n, k}, f) \leq \frac{k}{\max(0, k - q)}, \qquad (9)$$

*where*

$$q = \gamma \left\lceil \left( \log \frac{k}{\delta} \right) \middle/ \begin{cases} \log \dfrac{1 - \phi_{\max}^m}{\phi_{\max}^\tau (1 - \phi_{\max}^{m-\tau})} & \text{if } \phi < 1 \\ \log m - \log(m - \tau) & \text{if } \phi = 1 \end{cases} \right\rceil.$$

Observe that as $\phi_{\max} \to 0$ and the groups become increasingly homogenous, we recover the previously established bound of $k/(k - \gamma)$ in the setting where divided groups are cohesive (Thm. 2).[9]

# 6. Experiments

We empirically investigate the impact of ensuring JR through both real-world experiments and simulations.[10] In

the real-world setting we explore ranking comments on a collective response system (Konya et al., 2023b), where users shared their thoughts on campus protests and the right to assemble. The simulations investigate the implications of our theoretical findings based on the Mallows mixture model and can be found in Appendix D.2.

**The GreedyCC algorithm and its price.** Solving for the optimal JR set $S^*_{JR}$ is NP-hard in general (Bredereck et al., 2019; Elkind et al., 2022). For small instances, we may use an ILP formulation, but for large instances (e.g. on social media platforms), we would expect practitioners to resort to approximation algorithms. To understand the effects we might expect in practice, we too employ an approximation algorithm in all our experiments, along with an associated approximate price of JR. Specifically, we use the GreedyCC algorithm (Elkind et al., 2022; 2023) to find a set $S^{\text{Greedy}}_{JR}$ that satisfies JR and approximately maximizes the score function $f$. In brief, our implementation of GreedyCC greedily adds items to the selected set until a $n/k$-justifying set is found; the remaining items are then selected to maximize the score function $f$. A full description of the algorithm is given in Alg. D.1.

Given an instance $\mathcal{I}_{m, \mathcal{A}_n, k}$, we also define an associated price of GreedyCC based on comparing the score of the set $S^{\text{Greedy}}_{JR}$ returned by GreedyCC and the optimal set $S^*$ (which is not constrained to satisfy JR):

$$P^{\text{Greedy}}(\mathcal{I}_{m, \mathcal{A}_n, k}, f) = \frac{f(S^*(\mathcal{I}_{m, \mathcal{A}_n, k}, f))}{f(S^{\text{Greedy}}_{JR}(\mathcal{I}_{m, \mathcal{A}_n, k}, f))}. \qquad (10)$$

**Remesh dataset.** We now investigate the impact of enforcing JR in a real-world setting: ranking comments about campus protests and the right to assemble.[11] These comments were generated as part of two sessions on Remesh, a popularly-used *collective response system* (CRS) (Ovadya, 2023), each with about 300 participants (see Appendix E.1 for summary statistics about the dataset). Collective response systems are used by governments, companies, and non-profits to elicit the opinions of a target population at scale, in a more participatory way than traditional polling. In particular, on a CRS, participants give their opinion to different questions via free-form responses, and then vote on whether they agree with other participants' comments.[12]

**Experimental setup.** We examine the impact of ranking

---

[9]Note the ceiling in the bound.

[10]Our code builds upon the `abcvoting` Python package for approval-based multi-winner voting rules (Lackner et al., 2023).

[11]The data are available at `https://github.com/akonya/polarized-issues-data`.

[12]Individuals only give a feedback only on a small set of comments, however, we use inferences of the full approval matrix. In particular, we take the probabilistic agreement inferences (Konya et al., 2022) conducted by Remesh (in which each user and comment are given a probability of that user agreeing with that comment), and threshold these inferences by 0.5 to get a binary approval matrix.

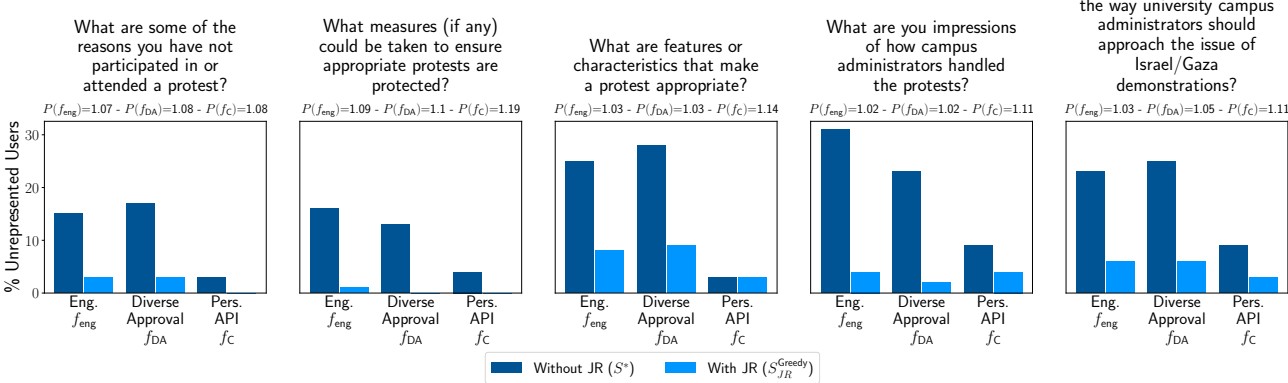

***Fig. 2***. **Representation when selecting** $k = 8$ **Remesh comments, with and without a JR constraint.** An individual who approves of at least one comment in a set $S$ is said to be *represented*. The JR constraint significantly increases the number of individuals who are represented. The prices of JR are reported with the shorthand $P(f)$ for $P^{\text{Greedy}}(\mathcal{I}_{m,\mathcal{A}_n},8,f)$. Due to space constraints, the figure only shows the five questions for which $S^*$ did not satisfy JR under diverse approval. Fig. E.2 shows results for the remaining five questions.

the comments using the following three scoring functions, with and without a JR constraint.

1. Engagement $f_{\text{eng}}$ (Eq. 2).

2. Maximin diverse approval $f_{\text{DA}}$ where participants' self-reported political ideologies were used to form the three groups (left-, center-, and right-leaning users) that were considered in the function $f_{\text{DA}}$ (see Eq. 3).

3. A Perspective API classifier $f_C$ that scores how "bridging" a comment is based upon its text (Saltz et al., 2024). The score of each comment is its average score for the seven "bridging" attributes in Perspective API: affinity, compassion, curiosity, nuance, personal story, reasoning, and respect.[13]

For each question in the Remesh dataset[14] and each score function $f$, we compare the optimal set $S^*(f)$ of $k = 8$ comments (without a JR constraint) and the set $S^{\text{Greedy}}_{JR}(f)$ which optimizes the score function while ensuring JR through the GreedyCC approximation algorithm (Alg. D.1).

### 6.1. Results

**Enforcing JR leads to more users being represented.** We say that an individual is represented in the set $S^*$ or $S^{\text{Greedy}}_{JR}$ if there is at least one item in the set that they approve of. For all ten questions and all three scoring functions, enforcing a JR constraint increases overall representation (Fig. 2). Without enforcing JR, 18%, 15%, and 4% of users were unrepresented in the engagement $S^*(f_{\text{eng}})$, diverse approval

$S^*(f_{\text{DA}})$, or the Perspective API set $S^*(f_C)$, respectively. After enforcing JR with GreedyCC, these percentages reduced to 5%, 4%, and 2%, respectively (Table E.4).

**Ranking with Perspective API always satisfied JR.** Without explicitly enforcing JR, the engagement $S^*(f_{\text{eng}})$, diverse approval $S^*(f_{\text{DA}})$, and the Perspective API set $S^*(f_C)$ satisfied JR 30%, 50%, and 100% of the time, respectively. This was surprising as the Perspective API does not explicitly aim to satisfy JR. Moreover, both diverse approval and the Perspective API classifiers are meant to "bridge" across groups (Ovadya & Thorburn, 2023). However, Perspective API satisfied JR at a much higher rate and also provided better overall representation (Fig. 2). This could be because diverse approval focuses on bridging across a few pre-defined groups (in this case, the three political groups), whereas the bridging attributes targeted by the Perspective API might appeal to a broader range of groups.[15]

**Enforcing JR improves representation across political groups.** As shown in Appendix E.6, enforcing JR with GreedyCC improves representation for all three political groups 100%, 100%, and 70% of the time, respectively (although in the case of Perspective API, the set already satisfied JR without GreedyCC). This is notable as the JR criterion does not rely on any explicit ideology labels of users.

**Enforcing JR comes at a low price.** We know that the price of JR $P(k, f_{\text{eng}})$ for engagement is in $\Theta(\sqrt{k})$ (Elkind et al.,

---

[13]See https://developers.perspectiveapi.com/s/about-the-api-attributes-and-languages for details on each attribute. We also report results broken down by individual attributes in Appendix E.6.

[14]We filter the data to remove any empty or duplicate comments.

[15]All our results rely on Remesh's inferred approval votes and biases in these inference could impact our results. For example, if the inferences are biased towards predicting approval for longer texts (we do not know if this is the case), which are also scored higher by the Perspective API, then the Perspective API might appear to be provide more representation than it actually does.

2022), for maximin diverse approval $f_{\mathrm{DA}}$ is equal to $k$ (Thm. 1), and for a general score function like the Perspective API may be unbounded (Thm. 1). Thus, in the worst case, we would expect that $P(k, f_{\mathrm{eng}}) < P(k, f_{\mathrm{DA}}) < P(k, f_C)$. We do indeed find this ordering when evaluating the price of GreedyCC, however, the prices are all also much lower than the worst-case bounds. Letting $\bar{P}^{\mathrm{Greedy}}(f)$ be the price of GreedyCC $P^{\mathrm{Greedy}}(\mathcal{I}_{m,\mathcal{A}_n,8}, f)$ averaged over the ten questions (instances) in the dataset, we found that:

$$\bar{P}^{\mathrm{Greedy}}(f_{\mathrm{eng}}) = 1.05$$
$$< \bar{P}^{\mathrm{Greedy}}(f_{\mathrm{DA}}) = 1.06$$
$$< \bar{P}^{\mathrm{Greedy}}(f_C) = 1.18.$$

These low prices may be explained by the presence of small $n/k$-justifying sets. For all questions, GreedyCC was able to find an $n/k$-justifying set with only at most $\gamma = 2$ comments (Table E.3), which from Theorem 2, implies that the price of JR can be at most $k/(k-\gamma) = 8/(8-2) = 1.33$.[16] A small $n/k$-justifying set would also be expected in settings where user opinions cluster into only a few groups (the same setting we analyze in Section 5).

**The JR feeds also satisfied EJR+.** Finally, out of the 48 JR feeds in our experiment, 47 of them also satisfied EJR+ (Brill & Peters, 2023), a strengthened version of JR. Although more experimentation is needed, these findings may indicate that for comment ranking, as in other real-world applications of JR (Boehmer et al., 2024), the practical differences between these axioms may be minimal.

## 7. Discussion

The online public sphere presents an opportunity to scale informal public deliberation. However, current tools for prosocial moderation and ranking of online comments may also inadvertently suppress legitimate viewpoints, undermining another key aspect of deliberation: inclusion of diverse viewpoints. In this paper, we introduced a general framework for algorithm ranking that incorporates a representation constraint from social choice theory. We showed, in theory and practice, how enforcing this JR constraint can result in greater representation while being compatible with optimizing for other measures of conversational quality or even user engagement. Our work sets the groundwork for more principled algorithmic interventions that can uphold conversational norms while maintaining representation.

**Limitations.** Nevertheless, there are still additional questions that should be considered before deploying our framework on a real-world platform.

First, the JR representation constraint depends on an ap-

proval matrix specifying which users approve which comments. It is not always clear which users should be included in the approval matrix. For example, should it be all users on the platform or only those who saw the post? The answer to this question may be different for different platforms.

Second, on real-world platforms, users will not "vote" on all comments, and thus, their approval of various comments will need to be inferred (Halpern et al., 2023). To provide users with genuine representation, it is essential to ensure these inferences are accurate. (Our Remesh experiments in Section 6 also relied on inferences of the approval matrix and should be interpreted with this in mind.) Moreover, on many platforms, users' approval will need to be inferred from engagement data such as upvotes or likes. If the chosen engagement significantly diverges from actual user approval, the validity of the process could be compromised. Investigating the impact of biased approval votes, in a similar vein to the work of Halpern et al. (2023) or Faliszewski et al. (2022), could be an interesting direction for future work

Finally, it is crucial to test the effects of enforcing JR through A/B tests, as offline results may differ from those observed in real-world deployments. In our Remesh experiments, we found that enforcing JR came at little cost to other conversational quality measures and also user engagement. However, these were "offline experiments" involving the re-ranking of historical data, without deploying our new algorithm to users, and further testing is still needed.

**Future work.** In this work, we focus on analyzing JR (although empirically, we find that all but one of our JR committees also satisfies EJR+). In future work, it would be interesting to analyze stronger axioms such as EJR (Aziz et al., 2017), EJR+ (Brill & Peters, 2023) or BJR (Fish et al., 2024). Additionally, our work focused on the classical JR setting, where a set of $k$ top comments is selected. However, it could be interesting to explore extensions of JR that apply directly to the ranking setting (Skowron et al., 2017; Israel & Brill, 2024), although it is not immediately clear how to integrate a scoring function into these extensions.

## Acknowledgements

We thank Hoda Heidari, Seth Lazar, Aviv Ovadya, Ariel Proccacia, Luke Thorburn, and Glen Weyl for their feedback on this project. We thank Andrew Konya and Lina Qiu for sharing the Remesh inferred approval votes with us and making them publicly available for future research.

## Impact Statement

Online comment sections have the potential to serve as important platforms for public discourse and deliberation, though their effectiveness is often undermined by the low quality of conversations. Algorithms can potentially sig-

---

[16]The proof of Theorem 2 follows from the fact that, if the population of users can be partitioned into $\gamma$ cohesive groups, then there is an $n/k$-justifying set of size $\gamma$.

nificantly improve these spaces by facilitating more constructive public sphere deliberation. This work takes a step in this direction by directly incorporating ideals of representation, as formalized in social choice theory, into the comment ranking process. We demonstrate that this approach can maintain conversational quality while ensuring diverse viewpoints are represented. Nonetheless, as discussed in our limitations section, before deploying this on a real-world platform, more testing is necessary to ensure that implementations of this approach genuinely provide users with representation.

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

## A. How does JR compare to demographic-based representation?

It is worth discussing how our JR-based approach differs from other, perhaps more familiar, notions of demographic or social representation from the algorithmic fairness literature (Barocas et al., 2023; Chasalow & Levy, 2021). JR is a bottom-up measure where the groups that are represented depend entirely on the users' own approval of different comments. This has several advantages for the comment ranking setting.

First, is the feasibility for real-world applications. In algorithmic fairness, groups are typically pre-defined based on a demographic, such as race or religion. In contrast to JR, the requirement of (inference of) user demographic labels make applying many algorithmic fairness methods infeasible or difficult to implement in industry settings, including on social media platforms, because of conflicts with privacy and legal constraints (Holstein et al., 2019; Veale & Binns, 2017).

Second, JR has the flexibility to automatically represent different groups for different settings. For example, the groups that are relevant to represent when ranking comments on an article about the 2025 New York City budget planning process are different from the groups that are relevant to represent on a article about a basketball game between the Celtics and Knicks. On a news site or social media platform where the relevent groups to represent for each post can be vastly different, it is difficult to scale up algorithmic fairness approaches that require pre-specifying the set of dimensions or demographics to consider.

Third, JR may accommodate intersectionality (Crenshaw, 1991) in a more natural way than many algorithmic fairness approaches. Algorithmic fairness approaches to intersectionality attempt to provide guarantees to a wide set of sub-groups (Gohar & Cheng, 2023), e.g., by defining intersectional groups as the combination of different demographic attributes (e.g. age, gender, race) (Kearns et al., 2018). However, this approach can lead to issues where the subgroups no longer correspond to meaningful entities. For instance, intersecting many dimensions can result in subgroups that are too specific and lack a meaningful reason for grouping (e.g., "Jewish white males aged 18-35 who have a bachelor's degree and live in a rural area"). In contrast, JR focuses on *cohesive* groups—groups of users who can agree on at least one comment in common—thereby inherently embedding *some* requirement for meaningfulness.

Finally, in cases where it is representation across pre-defined groups (e.g. based on demographic labels) that is important, these groups can still be considered through the score function $f$. We have already given one example, the diverse approval score function $f_{\text{DA}}$ defined in Equation (3), that uses explicitly defined groups in its computation.

# B. An Example of Conflict between Diverse Approval and JR

Here, we provide an example where minimax diverse approval $f_{DA}$ leads to sets that do not satisfy JR. The example is similar to that of Figure 1. There are two groups, $G_1$ and $G_2$, that are being bridged in the diverse approval objective function $f_{DA}$ (see Equation (3)). However, the only items that receive approval across both groups are because of two individuals $F$ and $G$ who approve of them (and no others). Thus, maximizing diverse approval leads to a set that $F$ and $G$ are represented in, but no one else.

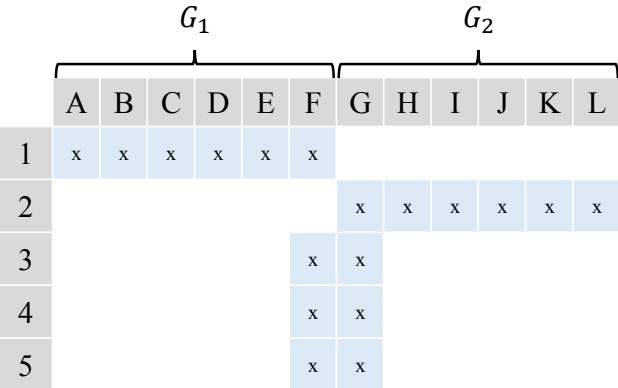

**Fig. B.1**. **When diverse approval fails at representativeness** An approval profile with $n = 12, m = 5$ and $k = 3$. There are $\gamma = 2$ cohesive groups $G_1 = \{A, B, C, D, E, F\}$ and $G_2 = \{G, H, I, J, K, L\}$ such that $|G_{1;2}| = 6 > n/k$. Note that the sub-groups such as $\{A, B, C, D\}$ and $\{I, J, K, L\}$ are also cohesive and larger than $n/k$, so they ought to also be represented by JR. In turn, any JR set $S$ contains $\{1, 2\}$. On the other hand, the comments' maximin diverse approval scores (across the groups $G_1$ and $G_2$) are $f(1) = f(2) = 0$ and $f(3) = f(4) = f(5) = 1/6$. In turn, the set with the highest score is $S^* = \{3, 4, 5\}$.

# C. Proofs

## C.1. Proof of Proposition 1

*Proof.* Let $m > k$ items and $\gamma = k$ groups each of size $n/k$ such that every group member in group $G_i$ approves of one item $s_i$ unique to the group. Each group $G_i$ is cohesive and of size $n/k$ so a JR set shall include all these unique items $s_i$ for all $i \in [\gamma]$. Recall that a general scoring rule (e.g. bridging) may be based on the output of a classifier whose input are the items (e.g. comments) and henceforth, be independent of the approval profile. In turn, we can construct an instance where the general scoring rule is such that $f(s_i) = \varepsilon$ for all these items and a larger constant $c$ for at least one of the remaining items. Then $f(S^*_{JR}) = k\varepsilon$ while $f(S^*) \geq c$ is a positive constant. We can make $\varepsilon$ arbitrarily small for an unbounded $P(k)$. $\square$

## C.2. Proof of Theorem 1

We first prove the following lemma. This statement can be found in (Bredereck et al., 2019) (Theorem 3), but the authors do not provide a proof for the result in their paper so we write one below.

**Lemma 1.** *If item $i^*$ is approved by one person, then there exists a JR set that contains $i^*$.*

*Proof.* Let us distinguish two cases. *Case 1:* There exists a set of size strictly less than $k$ that satisfies JR. Then, trivially, take an arbitrary set $S$ that satisfies JR. If item $i^*$ is not already in $S$, add $i^*$ (and complete with any remaining item if necessary to create a set of size $k$). This set of size $k$ satisfies JR.

*Case 2:* The only sets that satisfy JR are of size $k$. This happens if and only if $\exists$ a set $T$ of size $k$ such that $\forall i \in T$, $\exists$ a group of $n/k$ users $g = \{u_1, \cdots, u_{n/k}\}$ such that $i \in \cap_{u \in g} A_u$ and $i \notin A_{u'} \forall u' \in [n] \setminus g$. In other words, we are in Case 2 in there exists a set $T$ of size $k$ such that every item $t \in T$ is approved by exactly $n/k$ people why do not approve of any other item in $T$. Note that such $T$ satisfies JR.

Recall that $i^*$ is approved by at least one person who belongs to one of the $k$ groups $g$ of size $n/k$. That is, there exists a user group $g$ of size $n/k$ such that $u \in g$ and $i^* \in A_g$. We will denote by $i_g$ the item in $T$ that is approved by all users in $g$ and no users in $[n] \setminus g$.

If $i^* \notin T$, define $S = T \setminus \{i_g\} \cup \{i^*\}$. Then, $S$ is a set of size $k$ that satisfies JR given that $i^*$ represents the same group as $i_g$. $\square$

Let's next prove Theorem 1.

*Proof.* Let $i^*$ be the item with the highest score $s^*$. By additivity of the scoring rule, for any set $S$ of size $k$ and any approval profile $\mathcal{A}_n$, $f(S, \mathcal{A}_n) \leq ks^*$. In particular,

$$f(S^*, \mathcal{A}_n) \leq ks^*. \tag{11}$$

Next, by approval dependency of the scoring rule $f$, there must be at least one user who approves item $i^*$ (otherwise, its score is 0 as well as that of any other item and, by approval dependency, there does not exist any item approved by anyone, a trivial case for which the price is not defined). By 1, there exists a JR set $S_{JR}$ that contains item $i^*$. This may not be the optimal JR set $S^*_{JR}$, but, by definition of the optimal set, $f(S^*_{JR}, \mathcal{A}_n) \geq f(S_{JR}, \mathcal{A}_n)$.

Further note that, by additivity again,

$$f(S_{JR}, \mathcal{A}_n) \geq s^* \tag{12}$$

Combining Equations (11) and (12), we get that for any approval profile $\mathcal{A}_n$, $\frac{f(S_{JR}, \mathcal{A}_n)}{f(S^*_{JR}, \mathcal{A}_n)} \leq k$.

$\square$

## C.3. Proof of Theorem 2

*Proof.* First, we prove that $P(k) \leq \frac{k}{k-\gamma}$. Let $S^* = \{s_1, \ldots, s_k\}$ be the score maximizing $k$-set with respect to $f$, where the items are ordered by their scores such that $f(s_1) \geq f(s_2) \geq \cdots \geq f(s_k)$. Since each of the cohesive groups $G_1, \ldots, G_\gamma$ unanimously approves of at least one item, we can construct a set $R$ with $|R| \leq \gamma$, containing one unanimously approved item from each group. Since the groups $G_1, \ldots, G_\gamma$ partition the set of users, every user approves of at least one item in $R$. Therefore, any set of size $k$ which contains $R$ satisfies JR. The remaining $k - \gamma$ can be chosen to maximize the scoring rule

$f$. Consequently, there exists a set $S$ which satisfies JR and contains the $k - \gamma$ highest scoring items, $s_1, s_2, \ldots, s_{k-\gamma}$. This implies that $f(S)$ is at least $\frac{k-\gamma}{k} f(S^*)$, which shows that $P(k) \leq \frac{k}{k-\gamma}$.

We can adapt the above to show that the bound can be tight. Let each group be of equal size $\frac{n}{\gamma} > \frac{n}{k}$ and each group unanimously approve of exactly one distinct item so that $|R| = \gamma$ (in turn, users do not approve of any items other than their group's designated item). Note that every JR set must contain the set $R$. Therefore, if the items are such that for all $r \in R$, $f(r) = 0$ and for all $s \in S^*$, $f(s) = c$ for some positive constant $c$ (the top $k$ highest item scores are equal), then the maximum score for a JR set is $f(S^*_{JR}) = (k - \gamma)c$ while the optimal (non-JR) score is $f(S^*) = kc$. This establishes that $P(k) = \frac{k}{k-\gamma}$ for certain scoring functions.

We show that diverse approval is one such scoring function. Let $\gamma = \frac{nk}{n+k}$ (WLOG assume $n + k$ divides $nk$) groups with one consensus item per group (suppose it is item $x_i$ for group $i = 1, \ldots, \gamma$) approved only by the members of that group. Further, there is one unique member in each group that approves of an additional $k$ "shared" items (for concreteness suppose these items are $y_1, \ldots, y_k$; therefore each group has a member that approved of all these items). There are hence $m = \gamma + k$ items such that the first $\gamma$ items $x_1, \ldots, x_\gamma$ are approved by $n/k + 1$ persons within each group and have a diverse approval score of 0, and the last $k$ items $y_1, \ldots, y_k$ are approved by $\gamma$ persons with a diverse approval score of $\frac{k}{k+n}$. Each divided group contains a cohesive sub-group of size $n/k$ (these individuals all approve of item $x_i$) so a JR committee needs to include $x_1, \ldots, x_\gamma$. A set $S$ can become a JR set by completing it with $k - \gamma$ of the remaining items $y_1, \ldots, y_{k-\gamma}$. In turn, $f(S^*_{JR}) = (k - \gamma)\frac{k}{k+n}$ and $f(S^*) = k\frac{k}{k+n}$. □

### C.4. Proof of Theorem 3

There are several generative interpretations of the Mallows model (Lu & Boutilier, 2014) but one that is particularly useful for analyzing homogeneity is based on the Repeated Insertion Model (RIM) (Doignon et al., 2004). In our analysis, we make use of a modified version of RIM in which items from the reference ranking are inserted starting with the last ranked item and ending with the first ranked item. It can be shown this bottom up RIM sampling induces the same distribution as the usual top down RIM sampling.

---

**Bottom Up Sampling of Mallows** $\pi \sim M(\phi, \sigma)$

1. Let $\sigma_i$ denote the item ranked $i$-th in the reference ranking $\sigma$.
2. Initialize $\pi$ as an empty ranking.
3. Loop $i = m, m-1, \ldots, 1$:
    - Insert $\sigma_i$ into $\pi$ at rank position $j$ where $1 \leq j \leq m - i + 1$ with probability $\phi^{j-1}/(1 + \phi + \cdots + \phi^{m-i})$.

---

This generative version of Mallows gives us a nice way to express and bound the probability the top items are ranked below a given threshold.

**Lemma 2.** *Let $\pi \sim M(\phi, \sigma)$. Then for all $s \leq m$, the probability that none of the top $s$ items in the reference ranking $\sigma$ appear in the top $\tau$ items of $\pi$ is*

$$\Pr(\pi(1) > \tau, \pi(2) > \tau, \ldots, \pi(s) > \tau) \leq \begin{cases} \phi^{\tau s}(1 - \phi^{m-\tau})^s/(1 - \phi^m)^s, & \text{if } \phi < 1 \\ (1 - \tau/m)^s & \text{if } \phi = 1 \end{cases} \tag{13}$$

*Proof.* Let $\sigma_i$ be the $i$-th ranked item in reference ranking $\sigma$ and $\pi^{(i)}$ be the ranking obtained after inserting items $\sigma_m, \ldots, \sigma_i$. Therefore the final sampled ranking $\pi = \pi^{(1)}$. Let $S = \{\sigma_i : 1 \leq i \leq s\}$ and $U = \{\sigma_i : s + 1 \leq i \leq m\}$.

Case 1: $s \geq m - \tau$. Then there is not enough items in $U$ to fit in the top $s$ items of $\pi$ because $|U| \leq \tau$. Therefore an item from $S$ must be ranked in one of the top $s$ positions of $\pi$. Hence $\Pr(\forall i \leq s : \pi(i) > \tau) = 0$.

Case 2: $s < m - \tau$. We proceed by induction on $s$. Base case we have $s = 1$ and it is required that $\sigma_1$ be inserted at position $\tau + 1$ or below which occurs with probability $(\phi^\tau + \cdots + \phi^{m-1})/(1 + \phi + \cdots + \phi^{m-1})$.

Suppose Ineq. (13) holds for all $s$ up to some $s \geq 1$. Let $i = m, m-1, \ldots, 1$ be the loop iteration index as in the Bottom Up Sampling procedure and define $\ell = s - i + 1$. The first item from $S$ gets inserted when $\ell = 1$ (i.e. $i = s$). Let

$g_\ell = \min\{\pi^{s-\ell+1}(\sigma_r)|s-\ell+1 \le r \le s\}$ be the rank position of the highest ranked item from $S$ at iteration with index $i = s - \ell + 1$. We show $g_\ell$ is non-increasing in $\ell$. We can argue by induction on $\ell$. In base case when $\ell = 1$, $g_1$ is the position item $\sigma_s$ is in $\pi^{(s)}$. Suppose $g_r$ is non-increasing for all $r$ up to some $\ell$. Consider $\ell + 1$ when we insert $\sigma_{s-\ell+2}$, if it is inserted above position $g_\ell$ then $g_{\ell+1} < g_\ell$ and if it is inserted at or below $g_\ell$ then $g_{\ell+1} = g_\ell$. Therefore each item in $S$ must be inserted at position $\tau + 1$ or below with probability $(\phi^\tau + \cdots + \phi^{m-i})/(1 + \cdots + \phi^{m-i})$. Applied for all items in $S$ we multiply these insertion probabilities,

$$
\Pr(\forall i \le s, \pi(s) > \tau) = \prod_{i=1}^{s} \frac{\phi^\tau + \cdots + \phi^{m-i}}{1 + \cdots + \phi^{m-i}}
$$

$$
= \begin{cases} \displaystyle\prod_{i=1}^{s} \phi^\tau \frac{1 - \phi^{m-i-\tau+1}}{1 - \phi^{m-i+1}}, & \text{if } \phi < 1 \\[2ex] \displaystyle\prod_{i=1}^{s} \frac{m - i - \tau + 1}{m - i + 1} & \text{if } \phi = 1 \end{cases}
$$

$$
\le \begin{cases} \displaystyle\prod_{i=1}^{s} \phi^\tau \frac{1 - \phi^{m-\tau}}{1 - \phi^m}, & \text{if } \phi < 1 \\[2ex] \displaystyle\prod_{i=1}^{s} 1 - \tau/m, & \text{if } \phi = 1 \end{cases} \tag{14}
$$

where the inequalities are obtained when the quotient is maximized at $i = 1$. We get the upper bound in Ineq. (13) by noticing the terms Ineq. (14) no longer depends on $i$. $\qquad\square$

*Proof of Theorem 3.* We begin by showing that there exists an $n/k$-justifying set of size $q$ with probability at least $1 - \delta$. The bound on the price of justified representation is derived directly from this result. Consider the set $S = \sigma_1^{-1}([s]) \cup \sigma_2^{-1}([s]) \cup \cdots \cup \sigma_\gamma^{-1}([s])$ comprised of the top $s$ items in each group's reference ranking. The probability that there exists an $n/k$-justifying set of size $\gamma \cdot s$ is greater than or equal to the probability that the set $S$ is an $n/k$ justifying set, which we can lower bound as follows:

$$
\Pr(\exists S' \text{ s.t. } |S'| = \gamma s \text{ and } S' \text{ is an } n/k \text{ justifying set})
$$
$$
\ge \Pr(S \text{ is an } n/k \text{ justifying set})
$$
$$
\ge \Pr(|\{u : A_u \cap S = \emptyset\}| < n/k)
$$
$$
= 1 - \Pr(|\{u : A_u \cap S = \emptyset\}| \ge n/k)
$$
$$
= 1 - \Pr\left(\sum_{u=1}^{n} X_u \ge n/k\right)
$$
$$
\ge 1 - k \cdot \begin{cases} \phi_{\max}^{s\tau}(1 - \phi_{\max}^{m-\tau})^s/(1 - \phi_{\max}^m)^s, & \text{if } \phi < 1 \\ (1 - \tau/m)^s & \text{if } \phi = 1 \end{cases} \tag{15}
$$

where $X_u = 1\{u \mid A_u \cap S = \emptyset\}$ is the random variable indicating whether user $u$ has no item they approve of in the set $S$. By Lemma 2, we can bound $\Pr(X_u = 1)$ as in Ineq. (13). Thus, Ineq. (15) holds by Markov's inequality. When $\phi_{\max} < 1$, solving $1 - k \cdot \phi^{s\tau}(1 - \phi_{\max}^{m-\tau})^s/(1 - \phi_{\max}^m)^s \ge 1 - \delta$ for the parameter $s$, yields $s \ge \log(k/\delta)/\log((1 - \phi_{\max}^m)/(\phi_{\max}^\tau(1 - \phi_{\max}^{m-\tau})))$. In turn, with probability at least $1 - \delta$, there exists an $n/k$ justifying group of size $\gamma\lceil \log(k/\delta)/\log((1 - \phi_{\max}^m)/(\phi_{\max}^\tau(1 - \phi_{\max}^{m-\tau})))\rceil$. When $\phi_{\max} = 1$, the same calculation provides that, with probability at least $1 - \delta$, there exists an $n/k$ justifying group of size $\gamma\lceil \log(k/\delta)/\log(m/(m - \tau))\rceil$.

When $|S| \le k$, any set $W$ of size $k$ that contains $S$ satisfies justified representation. For the remaining items in $W$, we can pick the $\max(0, k - q)$ items with the highest bridging score. Thus, the highest bridging score $f(S_{JR}^*)$ attainable by a set that satisfies justified representation is at least $\frac{\max(0, k-d)}{k} f(S^*)$ where $S^*$ is the bridging-optimal set (unconstrained for justified representation). Therefore, we have the desired result that the price of justified representation is such that $P(\mathcal{I}_{m,\mathcal{A}_n,k}, f) \le k/\max(0, k - q)$. $\qquad\square$

# D. Simulations

In the following sections, we investigate the implications of our theoretical findings in simulations in the Mallows mixture model.

## D.1. The GreedyCC approximation algorithm

First, we review the GreedyCC algorithm used in both sets of experiments. Given that solving for the optimal JR set $S^*_{JR}$ is NP-hard in general (Bredereck et al., 2019; Elkind et al., 2022), we would expect practitioners to resort to approximation algorithms. Thus, to understand the effects we might expect in practice, in all our experiments, we too employ an approximation algorithm, along with an associated approximate price of JR. Specifically use the GreedyCC algorithm, which has also been used in prior work (Elkind et al., 2022), to find a set $S^{\text{Greedy}}_{JR}$ that satisfies JR and approximately maximizes the score function $f$. In brief, our implementation of GreedyCC greedily adds items to the selected set until a $n/k$-justifying set is found; the remaining items are then selected to maximize the score function $f$. A full description of the algorithm is outlined in Alg. D.1.

Given an instance $\mathcal{I}_{m,\mathcal{A}_n,k}$, we also define an associated price of GreedyCC based on comparing the conversational score of the set $S^{\text{Greedy}}_{JR}$ returned by GreedyCC and the optimal set $S^*$ (which is not constrained to satisfy JR):

$$P^{\text{Greedy}}(\mathcal{I}_{m,\mathcal{A}_n,k}, f) = \frac{f(S^*(\mathcal{I}_{m,\mathcal{A}_n,k}, f))}{f(S^{\text{Greedy}}_{JR}(\mathcal{I}_{m,\mathcal{A}_n,k}, f))} \,, \tag{16}$$

$$P^{\text{Greedy}}(k, f) = \max_{\mathcal{I}_{m,\mathcal{A}_n,k}} P^{\text{Greedy}}(\mathcal{I}_{m,\mathcal{A}_n,k}, f) \,. \tag{17}$$

---

**Algorithm D.1** The *GreedyCC* algorithm proceeds in two stages. In the first stage, it identifies an $n/k$-justifying set by greedily selecting comments to maximize coverage.[17] Once an $n/k$-justifying set is found, the algorithm proceeds to the second stage, where it fills any remaining slots with the comments with highest conversational norm scores.

---

1: Initialize $V = [n]$ as the set of all voters
2: Initialize $S \subseteq [m]$ as the set of all items that are approved by at least $n/k$ voters in $V$
3: Initialize $W = \emptyset$
4: **while** $|W| < k$ **do**
5:   **if** $|S| > 0$ **then**
6:     Add $i^* = \arg\max_{i \in S \setminus W} |\{u \in V \,:\, i \in A_u\}|$ to $W$ {Stage 1}
7:   **else**
8:     Add $i^* = \arg\max_{i \notin W} f(i, \mathcal{A}_n)$ to $W$ {Stage 2}
9:   **end if**
10:   $V \leftarrow$ the set of voters who do not yet approve of an item in $W$
11:   $S \leftarrow$ the set of all items that are approved by at least $n/k$ voters in $V$
12: **end while**
13: return $W$

---

## D.2. Mallows Mixture Model Simulations

In simulations with the Mallows model, we empirically investigate the price of GreedyCC and compare our findings to the theoretical bounds derived in Section 5.2. Our theoretical bounds showed that when the population is clustered, in the sense that the population can be divided into a few groups that exhibit high within-group homogeneity, then the price of JR can be shown to be low. However, is this still true when using an approximation algorithm like GreedyCC, which is what a practitioner would use in practice?

**Experimental setup.** To investigate this, we created a Mallows mixture model $\mathcal{M}_2([\phi, \phi], [\sigma_1, \sigma_2], [1/2, 1/2])$ that simulates a polarized environment with two groups. We draw inspiration from Esteban & Ray (1994)'s axiomatic framework for measuring polarization. Esteban & Ray (1994) where they consider polarization to be present if three intuitive properties are satisfied. We operationalized these three properties in our simulation as follows:

---

[17]The coverage of a set of comments $S$ is the total number of users who approve of any comment in $S$.

- *Criteria 1: There must be a small number of significantly sized groups.* The mixture model had two components and each component had a probability of $1/2$.

- *Criteria 2: There must be a high degree of heterogeneity across groups.* We chose the central rankings $\sigma_1$ and $\sigma_2$ of the two components to be exactly the opposite.

- *Criteria 3: There must be a high degree of homogeneity within groups.* Each component had the same dispersion paremeter $\phi$, and we varied this parameter to see the impact of wihtin-group homogeneity, which our theoretical results depended upon.

As in Section 5.2, we generate approval votes for a user's preference $\pi \sim \mathcal{M}_\gamma([\phi, \phi], [\sigma_1, \sigma_2], [1/2, 1/2])$ by thresholding and taking the top $\tau$ items to be approved. Based upon the approval vote thresholding, we let $I$ denote the random variable corresponding to an instance simulated from the Mallows mixture model, and denote by $\mathcal{D}$ the dataset of $s$ i.i.d. instances from the Mallows mixture model. We evaluated (1) the expected price of GreedyCC, over instances, estimated using the sample mean $\mathbb{E}_s[P^{\text{Greedy}}(I, f)]$, and (2) the worst-case price of GreedyCC observed in the dataset, $P_s^{\text{Greedy}}(k, f)$:

$$\mathbb{E}_s[P^{\text{Greedy}}(I, f)] = \sum_{\mathcal{I} \in \mathcal{D}} P^{\text{Greedy}}(\mathcal{I}, f)/s \,, \tag{18}$$

$$P_s^{\text{Greedy}}(k, f) = \max_{\mathcal{I} \in \mathcal{D}} P^{\text{Greedy}}(\mathcal{I}, f) \,. \tag{19}$$

**Results.**  Figure D.1 shows how the price of GreedyCC changes as the dispersion parameters $\phi$ changes, for two scoring functions: engagement $f_{\text{eng}}(i, \mathcal{A}_n)$ (Equation 2) and maximin diverse approval $f_{\text{MDA}}(i, \mathcal{A}_n)$ where the two groups in the definition of $f_{\text{MDA}}$ (Equation 3) are the two components in the Mallows mixture model.

First, despite GreedyCC not being optimal, we find that the price of GreedyCC is low (close to one–the lowest possible value) for both engagement and diverse approval when there is low dispersion, as predicted by our theoretical results. For both engagement and diverse approval, we find that as the dispersion increases, the price of GreedyCC increases until the dispersion is very high (at which point cohesive groups are unlikely to exist), and the price decreases again.

Second, we find that the price of GreedyCC is higher for diverse approval than for engagement. For diverse approval, some instances result in a price of GreedyCC that nearly our theoretical bound from Theorem 3. The gap in price between diverse approval and engagement is consistent with our result in Theorem 1, which found that the price of JR for maximin diverse approval $P(k, f_{\text{MDA}})$ equals $k$, whereas Elkind et al. (2022) demonstrated that the price for engagement $P(k, f_{\text{eng}})$ is $\Theta(\sqrt{k})$. These results were shown, in the general setting, without any restrictions on the approval profiles. On the other hand, the theoretical bound we derived for the polarized setting in Theorem 3, and plotted in Figure D.1, holds for all score functions. But given the empirical disparity in price between diverse approval and engagement, it seems plausible that a similar score-function-specific gap could be identified even in the polarized setting.

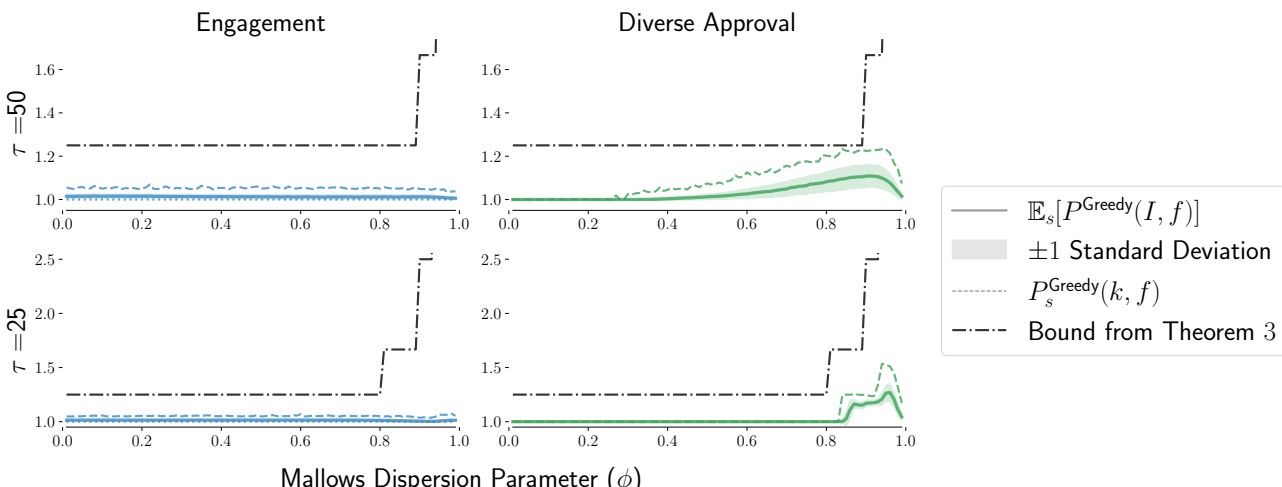

*Fig. D.1*. **The price of GreedyCC in a Mallows mixture model.** The price of GreedyCC is shown for both the engagement scoring rule $f_{\text{eng}}$ (left) and diverse approval $f_{\text{DA}}$ (right) with a $\tau = 25$ (top) and $\tau = 50$ (bottom). The simulations are for an instance $\mathcal{I}_{100, \mathcal{A}_{100}, 10}$ where the approval profile $\mathcal{A}_{100}$ is simulated through a Mallow mixture model with the Kendall-tau distance, $\gamma = 2$ polarized groups with opposite reference rankings, and an approval threshold of $\tau = 25$. The average price, $\mathbb{E}_s[P^{\text{Greedy}}(I)]$ (solid lines), and the maximum observed price $\overline{P_s}^{\text{Greedy}}(I)$ (dotted lines), are both computed over $s = 1,000$ simulations for values of $\phi$ in $[0.1, 1]$ and a 0.01-stepsize. The probabilistic bound from Theorem 3 (dash-dot black lines) is computed with a 95% confidence, i.e., $\delta = 0.05$.

# E. Experiments Ranking Comments on Remesh

In this section, we now provide supplementary details and results for our Remesh experiments from Sec. 6.

## E.1. Dataset Summary Statistics

For these experiments, we used data from Remesh sessions conducted in the summer of 2024 that engaged a representative sample of Americans regarding their opinion of campus protests. The data are available at `https://github.com/akonya/polarized-issues-data`. The following table lists the 10 questions that participants were asked about, across the two sessions, and the total number of participants and comments for each question.

| ID | Questions | Participants ($n$) | Comments ($m$) |
|----|-----------|-------------------|----------------|
| 1 | What are your thoughts on the way university campus administrators should approach the issue of Israel/Gaza demonstrations? | 307 | 306 |
| 2 | What are your impressions of how campus administrators handled the protests? | 308 | 308 |
| 3 | What should guide university campus administrators handling of protests? | 301 | 298 |
| 4 | What are your impressions of the campus protests? | 310 | 310 |
| 5 | How has your personal experience with protests influenced your viewpoint on the right to assemble? | 105 | 103 |
| 6 | What characteristics or actions, in your view, deem a protest inappropriate? | 306 | 297 |
| 7 | What are some of the reasons you have not participated in or attended a protest? | 201 | 200 |
| 8 | What are features or characteristics that make a protest appropriate? | 307 | 306 |
| 9 | What measures (if any) could be taken to ensure appropriate protests are protected? | 305 | 297 |
| 10 | What measures (if any) could be taken to restrict or limit inappropriate protests? | 303 | 284 |

*Table E.1.* **Remesh Survey Questions and Number of Participants ($n$) and Comments ($m$)**

## E.2. Perspective API Classifiers

We used three scoring functions to rank comments: engagement $f_{\text{eng}}$, diverse approval $f_{\text{DA}}$, and a bridging score based on Google Jigsaw's Perspective API $f_C$. For the Perspective API score function $f_C$, we used the average of the scores for the seven "bridging" attributes available in Perspective API (see `https://developers.PerspectiveAPI.com/s/about-the-api-attributes-and-languages?language=en_US`): Nuance, Compassion, Personal Story, Reasoning, Curiosity, Affinity, Respect. Below, we show the score distribution of the Remesh comments, for each attribute.

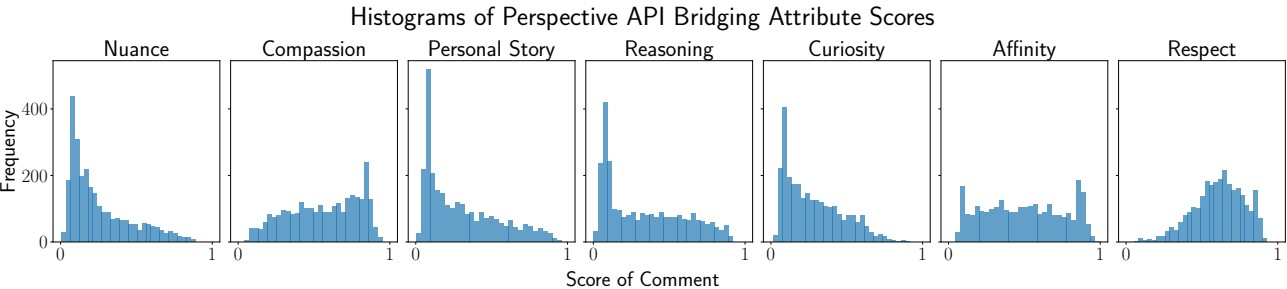

*Fig. E.1.* **Scores from the Perspective API**

For our main results reported in Figure 2, Table E.2, Table E.4 and Figure E.2, we use the average score across these seven attributes as the Perspective API score function $f_C$. However, we also report results with each individual attribute in

Appendix E.6.

### E.3. Price of GreedyCC when Ranking Comments using Diverse Approval

In Table E.2, we provide the price of GreedyCC $P^{\text{Greedy}}(\mathcal{I}_{m,\mathcal{A}_n,8}, f)$ for each question (instance $\mathcal{I}$) and score function $f$. We also indicate how often the optimal set $S^*(f)$ (which was not explicitly constrained to satisfy JR) satisfied JR.

| | Engagement | | Diverse Approval | | Perspective API Index | |
|---|---|---|---|---|---|---|
| Questions | $P^{\text{Greedy}}(\mathcal{I}, f_{\text{eng}})$ | $f_{\text{eng}}$ feed is JR | $P^{\text{Greedy}}(\mathcal{I}, f_{\text{DA}})$ | $f_{\text{DA}}$ feed is JR | $P^{\text{Greedy}}(\mathcal{I}, f_C)$ | $f_C$ feed is JR |
| 1 | 1.03 | False | 1.05 | False | 1.11 | True |
| 2 | 1.02 | False | 1.02 | True | 1.11 | True |
| 3 | 1.00 | True | 1.00 | True | 1.06 | True |
| 4 | 1.03 | False | 1.03 | False | 1.11 | True |
| 5 | 1.09 | False | 1.08 | True | 1.07 | True |
| 6 | 1.11 | True | 1.11 | True | 1.15 | True |
| 7 | 1.06 | False | 1.08 | False | 1.08 | True |
| 8 | 1.03 | False | 1.03 | False | 1.14 | True |
| 9 | 1.07 | False | 1.10 | False | 1.19 | True |
| 10 | 1.07 | True | 1.08 | True | 1.16 | True |

*Table E.2.* **Price of GreedyCC and frequency of satisfying JR for each score function and question.**

### E.4. Size of the $n/k$-justifying Set Found by GreedyCC

In Section 5.2, we show that when the user population clusters into a few groups, then the price of JR will be low because it is possible to find a small $n/k$-justifying set (and thus, the remaining items for the set can be chosen to maximize the score function). Indeed, this logic is how the GreedyCC approximation algorithm that we use operates. As detailed in Algorithm D.1, the GreedyCC proceeds in two stages. In the first stage, it finds an $n/k$-justifying set. In the second stage, it selects the remaining comments that maximize the score function. Thus, the smaller the $n/k$-justifying set, the lower the price of GreedyCC will tend to be.

A small $n/k$-justifying set is expected in empirical settings where user opinions cluster into only a few groups (as is common for politically-charged questions). Table E.3 shows the size of the $n/k$-justifying set found by GreedyCC for all ten questions in the Remesh dataset. For all questions, the $n/k$-justifying set has at most only two comments. The small size of the $n/k$-justifying sets could then explain why the price of GreedyCC (Table E.2) across all questions and score functions is so low (close to one in all cases).

| Questions | $n/k-$**justifying set** |
|---|---|
| 1 | 2 |
| 2 | 2 |
| 3 | 1 |
| 4 | 2 |
| 5 | 2 |
| 6 | 2 |
| 7 | 2 |
| 8 | 2 |
| 9 | 2 |
| 10 | 2 |

*Table E.3.* **The size of the $n/k-$justifying set found by GreedyCC for each question.**

### E.5. The Impact of JR on Representation of Participants

In Figure 2, we showed the proportion of participants who were unrepresented when ranking Remesh comments both with and without a JR constraint. Due to space constraints, we only included the five Remesh questions for which $S^*(f_{DA})$ did not satisfy JR in Figure 2 under diverse approval. Here, we present the results for the remaining five questions (Table E.4 and Figure E.2).

| Questions | Engagement | | Diverse Approval | | Perspective API Index | |
|---|---|---|---|---|---|---|
| | $S^*$ | $S_{JR}^{\textbf{Greedy}}$ | $S^*$ | $S_{JR}^{\textbf{Greedy}}$ | $S^*$ | $S_{JR}^{\textbf{Greedy}}$ |
| 1 | 23 | 6 | 25 | 6 | 9 | 3 |
| 2 | 31 | 4 | 23 | 2 | 9 | 4 |
| 3 | 8 | 8 | 5 | 5 | 0 | 1 |
| 4 | 22 | 10 | 15 | 7 | 9 | 3 |
| 5 | 16 | 1 | 8 | 2 | 6 | 2 |
| 6 | 10 | 0 | 10 | 0 | 5 | 0 |
| 7 | 14 | 4 | 17 | 3 | 3 | 0 |
| 8 | 25 | 8 | 28 | 9 | 3 | 3 |
| 9 | 15 | 3 | 13 | 0 | 4 | 0 |
| 10 | 12 | 1 | 10 | 2 | 1 | 1 |
| Average across all feeds | **18** | **5** | **15** | **4** | **5** | **2** |
| Average across the five feeds in Figure 2 | **22** | **4** | **21** | **4** | **6** | **2** |
| Average across the five feeds in Figure E.2 | **13** | **5** | **10** | **3** | **4** | **1** |

*Table E.4.* **The percentage of users who were unrepresented when ranking with each score function, both with the JR constraint ($S_{JR}^{\textbf{Greedy}}$) and without the JR constraint ($S^*$)**

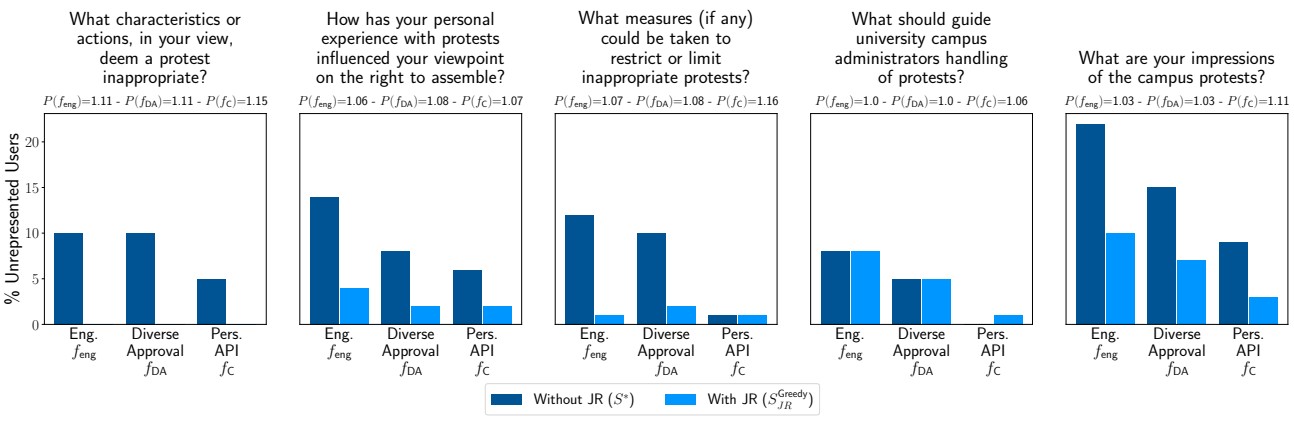

*Fig. E.2.* **Representation when selecting comments using diverse approval with and without a JR constraint (missing feeds).**

In Figure E.3, we plot the average score per comment for the set that maximizes the scoring function $f$ as a function of the percentage of users represented by the selected set. This is shown for both the unconstrained optimal set $S^*(f)$ and the set $S_{JR}^{\text{Greedy}}(f)$ that satisfies JR and approximately maximizes the score function $f$. We effectively calculate $\frac{f(S^*(f))}{k}$ and $\frac{f(S_{JR}^{\text{Greedy}}(f))}{k}$ respectively.[18]

### E.6. Results Per Political Ideology

In this section, we report the representativeness results across the different political groups for the three scoring functions: engagement $f_{\text{eng}}$, diverse approval $f_{\text{DA}}$, and the Perspective API bridging classifier $f_C$.

---

[18]We acknowledge an anonymous reviewer for suggesting this plot.

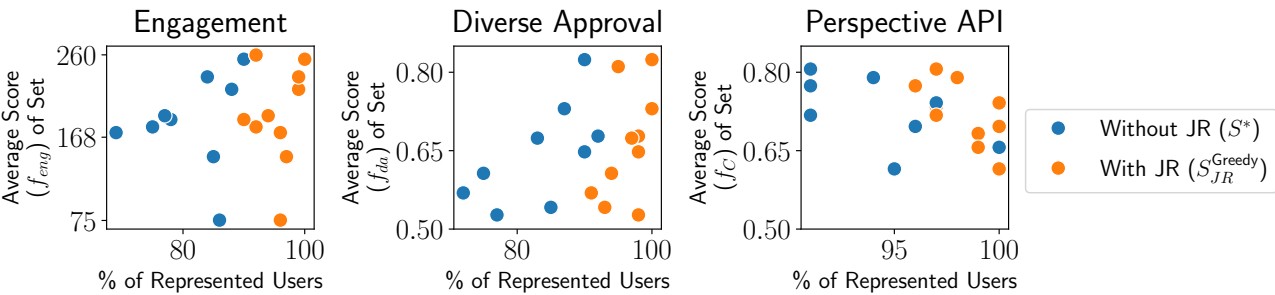

**Fig. E.3.** **The score-optimal sets $S^*$ and the JR sets $S_{JR}^{\textbf{Greedy}}$ for each Remesh question (see Table E.1) and score function (engagement, diverse approval, and the Perspective API bridging score).** Each set is plotted based on the average score of the items in the set and the percentage of users represented in the set. Compared to the score-optimal sets $S^*$, the JR sets $S_{JR}^{\text{Greedy}}$ notably increase the percentage of represented users without significantly reducing the score of the selected items.

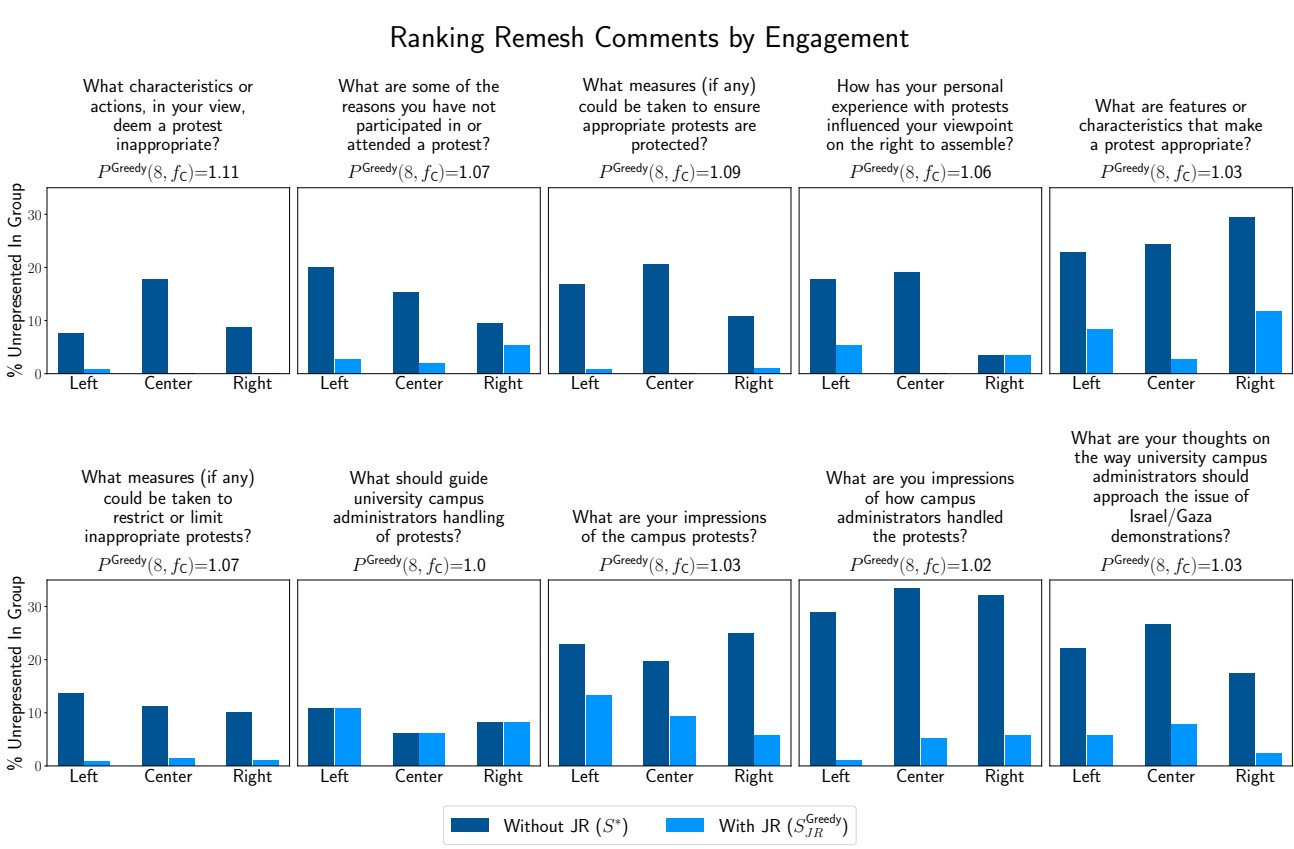

**Fig. E.4.** **Representation of political groups when ranking with the diverse approval $f_{\textbf{eng}}$**

## Ranking Remesh Comments by Diverse Approval

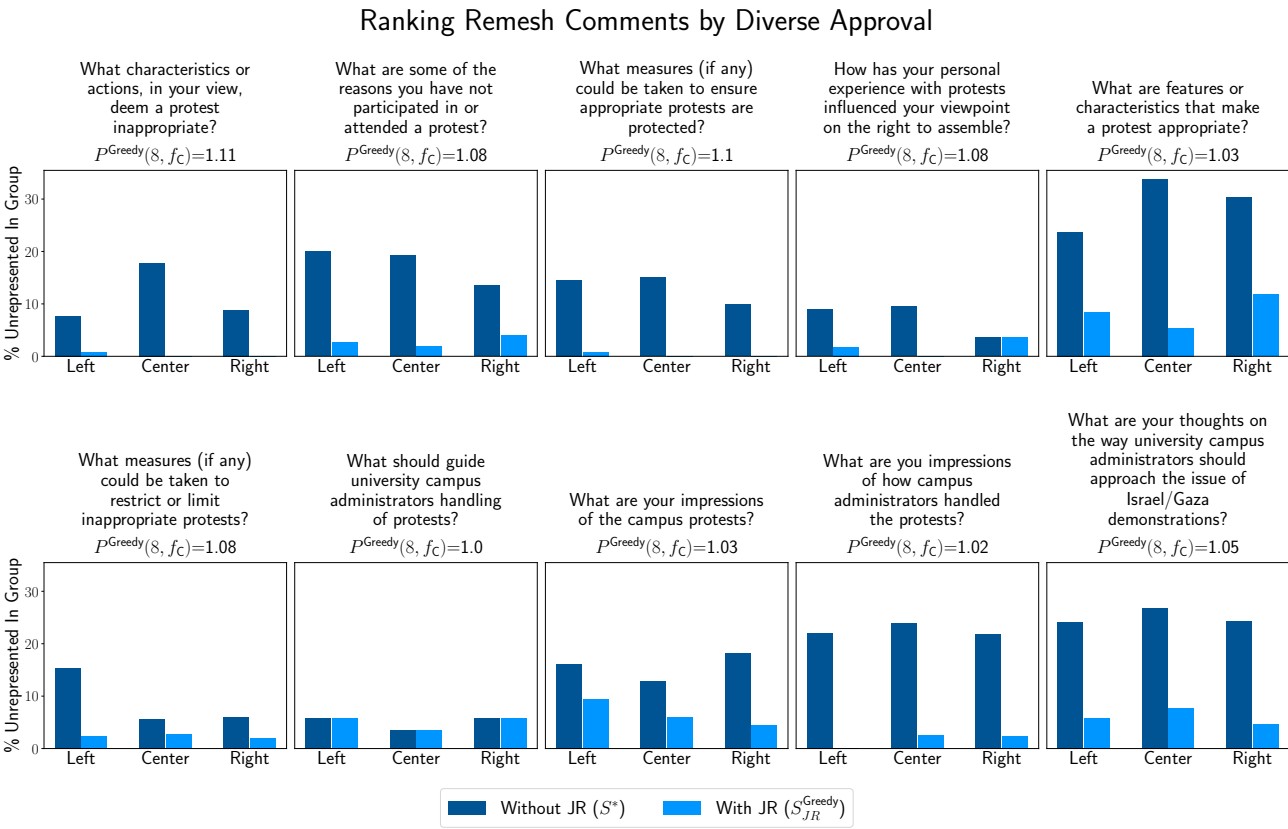

**Fig. E.5**. **Representation of political groups when ranking with the diverse approval $f_{\text{DA}}$**

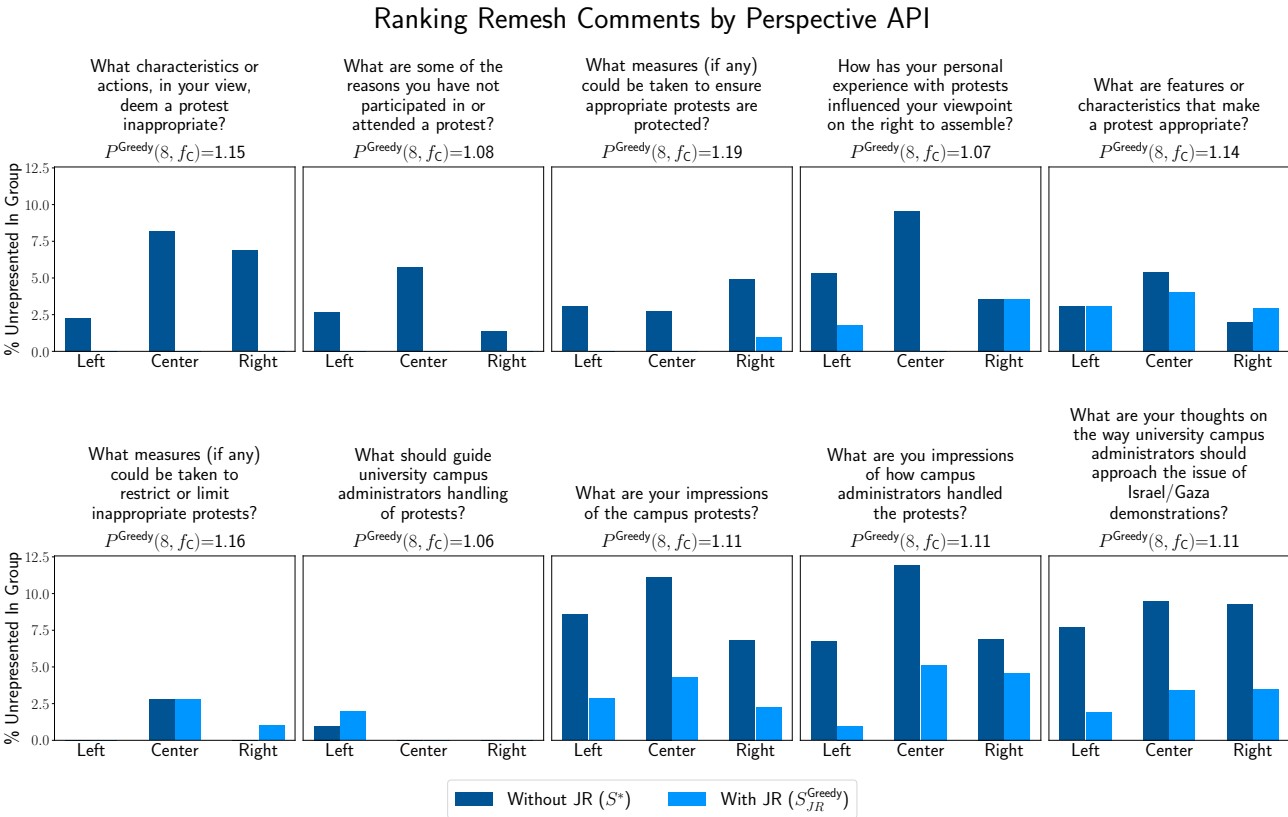

**Fig. E.6.** **Representation of political groups when ranking with the Perspective API** $f_C$

## E.7. Individual Attributes from Perspective API

In Figure E.7, we show the representation of users and the price of GreedyCC when ranking with each of the seven individual Perspective API attributes: nuance, compassion, personal story, reasoning, curiosity, affinity, and respect.

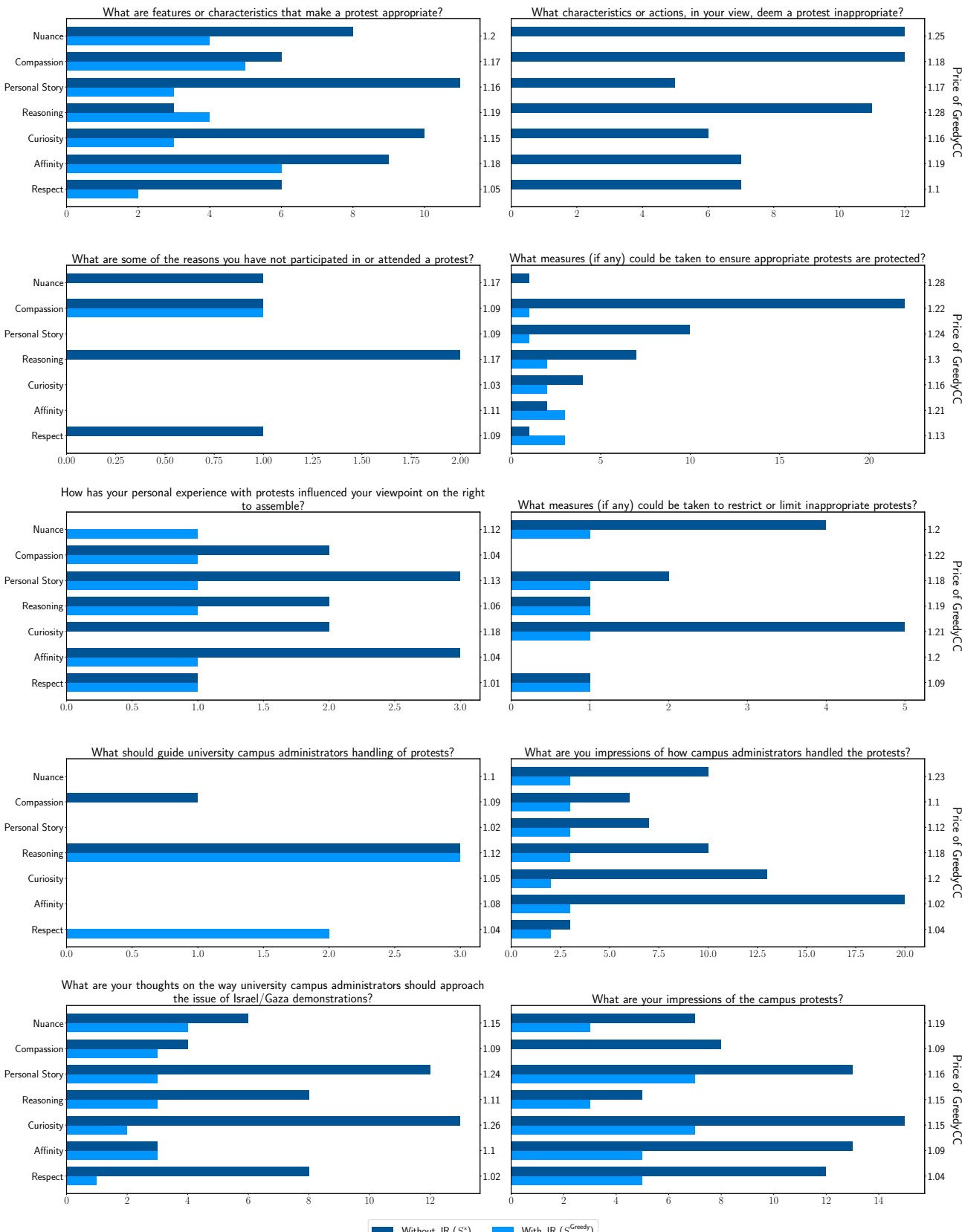

***Fig. E.7.* Representation of users when ranking with individual Perspective API attributes, both with the JR constraint ($S_{JR}^{\mathbf{Greedy}}$) and without the JR constraint ($S^*$).**

**E.8. The Difference in Comments Selected With and Without JR**

Lastly, we show an example of the feeds chosen by the three scoring functions; engagement $f_{\text{eng}}$, diverse approval $f_{\text{DA}}$, and the Perspective API $f_C$; for the question **"What measures (if any) could be taken to ensure appropriate protests are protected?"** Overall, the comments chosen by the Perspective API are much longer and more substantial than the comments chosen by engagement or diverse approval.

| | $S_{JR}^{\text{Greedy}}(f_{\text{eng}})$ | |
| | $S^*(f_{\text{eng}})$ | |
| **Extra comments picked by GreedyCC** | **Highest scoring comments** | **Next-highest scoring comments** |
| --- | --- | --- |
| Crowd control; dialogue. | Body search and metal detectors for a specific designated assembly location and strong police presence [...]. | More of a police presence. |
| | There a no inappropriate protests, only inappropriate protesters. | |
| | I think punishing those who make them inappropriate would help. | |
| | Put harsher punishments for if people do them, fines, jail time, other things. | |
| | This is difficult to answer.[...] Heightened police presence should help deter violence though. | |
| | Not issuing permits and leaders of the protest publicly denouncing. | |
| | Police force to shut these down. | |

*Table E.5.* **The engagement-maximizing set** $S^*(f_{\text{eng}})$ **and the JR-constrained engagement set** $S_{JR}^{\text{Greedy}}(f_{\text{eng}})$ **for the question "What measures (if any) could be taken to restrict or limit inappropriate protests?" .**

| | $S_{JR}^{\text{Greedy}}(f_{\text{DA}})$ | |
| --- | --- | --- |
| | | $S^*(f_{\text{DA}})$ |
| **Extra comments picked by GreedyCC** | **Highest DA comments** | **Next-highest DA comments** |
| Body search and metal detectors for a specific designated assembly location and strong police presence [...]. | No one really wants this. Again, it's about people being smart. Without that, we're just spitting in the wind. | Make rules and regulations stronger |
| Crowd control; dialogue. | I think punishing those who make them inappropriate would help. | Government approval of the protest. |
| | This is difficult to answer. [...] Heightened police presence should help deter violence though. | |
| | Depends. | |
| | This is a ridiculous question. I reject the idea that there is such a thing as an "inappropriate protest" at a conceptual level. The premise of the question is flawed and patently absurd. | |
| | Proper police presence and good supervision to ensure existing laws are not broken nor selectively enforced. | |

*Table E.6.* **The diverse-approval-maximizing set $S^*(f_{\text{DA}})$ and the JR-constrained diverse approval set $S_{JR}^{\text{Greedy}}(f_{\text{DA}})$ for the question "What measures (if any) could be taken to restrict or limit inappropriate protests?"**

| | $S_{JR}^{\text{Greedy}}(f_C)$ | |
| | $S^*(f_C)$ | |
| **Extra comments picked by GreedyCC** | **Highest scoring comments** | **Next-highest scoring comments** |
| --- | --- | --- |
| Body search and metal detectors for a specific designated assembly location and strong police presence [...]. | I don't think we should limit any protests, the right to assemble must be protected at all costs. [I]t is important that we keep the playing feild level in case we are ever on a side deemed "wrong" we wouldn't want to be denied that right. | I don't know these days. People are willing to do anything. It would be nice if no one was violent, but if you lived where I do, you would give up (as I have.) |
| Crowd control; dialogue. | Again, I think this goes to broader cultural issues. We hold fast to our "right" to protest. However, I think a protest could be restricted or limited if there are already plans for harms against others such as the group having a message of hate or postings about planning violence, of any type, on social media. | [...] Not having fake news spread over social media and people be less emotional before protesting, level-headed, and respectful. Consequences for violent and hateful actions by the protesters as if they are doing things that are going against the law, its their job to then deal with their consequences and respect police. Notify police before and see if they are available. |
| | In this democracy, it is difficult to to even determine what kind of protest is "inappropriate" unless the definition is destruction of property, violence, rioting, and that kind of thing. I personally do not believe in absolute freedom of speech, and in particularly feel that Nazi/white supermacist protests are always out of line. [...] | |
| | I think a greater oversight of protest permits is needed, and it should be okay to limit people from protesting if they have a history of violent behavior in previous protests. Free speech is okay but using it as an excuse for violence and destruction is not. | |
| | Unfortunately there aren't many steps to be taken. Bad apples are always around and use the protests as a means to create chaos and act inappropriately by being a thief, vandal, assaulter, etc. Sometimes these people are even planted by those opposing the protest to diminish the validity of the original cause. Having protests during daylight hours might help some. | |
| | I think people are breaking loss. They should be arrested. However, sometimes there is an overstepping as you can see in other countries where people are just arrested for protesting. | |

*Table E.7.* **The Perspective API set** $S^*(f_C)$ **and the JR-constrained Perspective API set** $S_{JR}^{\text{Greedy}}(f_C)$ **for the question "What measures (if any) could be taken to restrict or limit inappropriate protests?"**

