# OpenReview forum: "Representative Ranking for Deliberation in the Public Sphere"
_ICML.cc/2025/Conference — ICML 2025 poster_

### Official Review · Reviewer_q4GL · 2025-03-10

**Overall Recommendation:** 3

**Summary:**

The paper studies a setting of algorithmic comment ranking/selection incorporating fairness. There is a given set of comments, together with "likes" and based on these likes, a representative set of comments needs to be selected. The paper studies the impact of a group fairness concept called "justified representation" (JR) from the computational social choice literature. While in the worst-case, JR might inhibit the maximization of objective functions, in experiments on real-world comment data, as well as restricted settings, JR is compatible with approximately optimal function maximization.

**Claims And Evidence:**

They seem both clear and convincing.

**Essential References Not Discussed:**

I do not believe so

**Experimental Designs Or Analyses:**

Yes, checked them

**Methods And Evaluation Criteria:**

Yes, I would say so.

**Other Comments Or Suggestions:**

Line 94: e.g,
Line 104: For cultural reasons, I think using such country specific examples should be avoided. No one outside of the US would understand this example.
Related Work: The related work section is currently missing computational social choice works entirely, even though the paper is on computational social choice. This seems wrong to me.
Line 152: \cdots -> \dots (also elsewhere)
Line 143 (right): \cup -> \bigcup (also elsewhere)
Line 199 (left): the use of quantors in the text is quite ugly
Line 195 (right): \subset -> \subseteq (also elsewhere)
"It appears as though there could be perverse sets that satisfy JR where
most people do not approve of the selected comments. However,
research has found that common algorithms used to
satisfy JR do not lead to such perverse outcomes" Yes, common algorithms selecting JR outcomes are usually good. However, this does not mean that all JR outcomes are usually good (which might also be quite relevant here)
Theorem 4.2: \gamma is never used in the theorem statement
Theorem 5.1: I was wondering if you cant replace this whole construction by requiring that there is a small justifying set? I believe this is the only thing you need for the theorem.
I find Figure 2 quite overwhelming at the moment. I believe it could be a bit simplified or spread out.
You are citing the Bredereck et al paper twice.
The Elkind et al paper has a journal version
"W¨”uthrich" -> "Wüthrich"
"Landemore, H. 39Can Artificial Intelligence Bring Deliberation
to the Masses?" Cut the 39

While searching for related work I came across the paper "Combining Voting and Abstract Argumentation
to Understand Online Discussions" by Bernreiter et al., they seem to have a quite similar motivation to this work, but the results are different. Might still be worth to cite them though.

You are currently missing an impact statement. I believe this paper might actually be one that needs one.

**Other Strengths And Weaknesses:**

In general, I like the idea of the paper. Applying social choice fairness concepts to machine learning problems is a very interesting topic, and of growing importance in the last few years. The problem of comment ranking seems also quite important and well motivated.

There are a few things I am not too satisfied with, though:
(i) firstly, the whole paper builds upon the notion of JR. However, JR is an incredibly weak axiom. As the authors note themselves, in a lot of their experiments 2 comments are already enough to satisfy JR (this is inline with other experimental works in social choice, see for instance the cited Bredereck et al. paper). Further, JR does not meet the intuitive notion of proportionality suggested by the paper itself ("Let a world with 60 people who approve the 10 items in set A and another 40 people
who approve of a distinct set A↑ of 10 items. If a committee of size 10 is selected based on approval scores, the winning committee,
A would fail to represent 40% of the world. Instead, a committee composed of 6 items from A and 4 items from A↑ would respect
an intuitive notion of proportionality.") In this example, JR would only require 1 item to be selected from each group (in the computational social literature the property the paper suggests here is also known as "lower quota"). There are significantly stronger yet still intuitive fairness axioms than JR, which could have been used instead, for instance priceability[1] or EJR+[2] (I would also encourage the authors to look at the recent work of Boehmer et al. [3] on proportional representation in a real-world approval-based committee voting setting). I believe using such stronger axioms would significantly improve the quality of the experiments and results.
(ii) The theoretical results sadly seem quite weak. It is already known from the literature that the price of JR is \sqrt(k), so quite bad. It is not really surprising that for arbitrary functions this gets worse. Further, Theorem 5.1 is also not really that new, see for instance Proposition 3 of Lackner and Skowron [4].
(iii) One thing I found quite confusing is that the paper styles itself as being about ranking comments. The paper itself, however, is only about set selection, and I am not sure the results would entirely transfer to the ranking setting. In particular, Theorem 5.1 needs the set structure, if I see correctly.


References
[1] Proportionality and the Limits of Welfarism. Dominik Peters and Piotr Skowron 2020
[2] Robust and verifiable proportionality axioms for multiwinner voting. Markus Brill and Jannik Peters 2023
[3] Approval-based committee voting in practice: a case study of (over-) representation in the Polkadot blockchain. Niclas Boehmer et al. 2024
[4] Utilitarian welfare and representation guarantees of approval-based multiwinner rules. Martin Lackner and Piotr Skowron 2020.

**Questions For Authors:**

Can Theorem 5.1 be rephrased using justifying sets?

Why does the paper focus on JR and not stronger axioms?

**Relation To Broader Scientific Literature:**

The paper builds upon the concept of justified representation from computational social choice. This paper provides a novel application for justified representation and provides a few new theoretical results. In particular, the problem of maximizing an arbitrary function subject to JR was not really studied before. In general, the theoretical ideas  and results in the paper are quite similar to already present stuff in the computational social choice literature.

**Theoretical Claims:**

I did for the most part. Did not fully check the Mallows stuff.

---

> ### Author Rebuttal · Authors · 2025-03-31
>
> We thank the reviewer for their comprehensive and thoughtful comments.
>
> The reviewer seems to be evaluating our paper primarily as a contribution to social choice. However, the main goal of our work was not to contribute to social choice, but rather to facilitate deliberation online (particularly on social media platforms). Since this area is lacking any representation axioms, we ground our algorithmic approach in social choice, specifically JR, and demonstrate the efficacy of this approach both theoretically and empirically under real-world conditions. We believe this is an important and novel contribution that can open up new directions in the research on prosocial recommender systems. This goal explains many of the decisions we made, and we will try to make this context clearer in our revision.
>
> **Our focus on JR**
>
> The reviewer asked why we did not focus on stronger variants of JR. Practically speaking, a representation constraint is unlikely to be implemented on a social media platform if it also comes at a substantial decrease in user engagement or a substantial increase in the toxicity of comments. This is why in this initial work we started with the basic JR axiom, which is most likely to be compatible with the optimization of other score functions (engagement, civility classifiers, etc).
>
> However, in response to the reviewer’s concern, we also checked how many of the JR committees in our experiments satisfied EJR+ (and will add this to the paper). Across the 48 JR feeds, only one JR feed did not also satisfy EJR+. The paper that the reviewer mentioned by Boehmer et al. makes similar observations (e.g., "All four proportional rules [including three rules not guaranteed to satisfy EJR+] return committees satisfying EJR+ for all tested instances.’’).
>
> Overall, we believe focusing on JR was a reasonable first step in our work due to (i) the need to demonstrate that the representation axiom is compatible with other arbitrary score functions, and (ii) the observed alignment of JR and EJR+ in real-world settings despite the theoretical gap. Nonetheless, we very much agree with the reviewer that studying stronger axioms would be a promising direction for future research.
>
> **Theoretical results**
>
> We respectfully disagree with the reader’s assessment of our theoretical results. Coming back to our motivation of integrating JR into online platforms, the goal of our theoretical analysis was to show that, in natural settings, JR is compatible with the optimization of other score functions the platform may care about. While we agree that it is not surprising that the worst-care price of JR becomes higher for arbitrary score functions, these worst-case results primarily serve as motivation for our more novel contribution where we show that, in natural settings, the price is still low (Section 5). This result is crucial, as high prices would likely deter the incorporation of representation constraints on these platforms.
>
> Theorem 5.1 could be re-written in terms of n/k-justifying sets—that is essentially the proof of the theorem. However, our goal was to show a natural condition for why one might expect a small n/k-justifying set, and thus a low price of JR, on social media. In the social media context, when bridging interventions (diverse approval) are done, users are already partitioned into non-overlapping groups that are supposed to be distinctive in their preferences (these groups are typically learned by basically clustering the vote matrix). Theorem 5.1 says that if those groups are also cohesive in the JR sense, then we have a low price. We will clarify this reasoning and motivation, while acknowledging the prior related results, in our revised text.
>
> We also note that, in practice, each group may not necessarily be fully cohesive in the JR sense. Thus, Theorem 5.1 was also primarily meant to serve as intuition for the extended Mallows model result (Theorem 5.4).
>
> **Set selection and ranking**
>
> The reviewer is puzzled about the focus on set selection for comment ranking. We do not see a discrepancy here. In recommender systems, a very common method for showing a diverse set of items (e.g. videos that span your interests) is to re-rank items greedily so that the set of top items is diverse (see Algorithm 1 in “Fairness and Diversity in Recommender Systems: A Survey” by Zhao et al). Moreover, real-world recommender systems proceed in multiple steps that are actually set selection problems (e.g. retrieval -> early stage-ranking -> late-stage ranking). In our revision, we will clarify the connection between set selection and ranking.
>
> **Related Work & Impact Statement**
>
> We appreciate the many references that the reviewer shared. We will incorporate them into our paper with a more comprehensive section on related work in social choice. We will also add an impact statement. Lastly, we will correct the typos the reviewer found.
>
> Once again, we thank the reviewer for their thoughtful consideration of our paper.

---

> > ### Comment · Reviewer_q4GL · 2025-04-03
> >
> > Thank you for the nice rebuttal!
> >
> > Yes, it is correct, I tried to evaluate the paper from a social choice perspective, I guess that is what happens with such an "interdisciplinary" paper. I still maintain my position regarding the strength of the paper as a pure social choice work, namely, that in this regard it is quite thin content-wise, and that the presented content could be substantially improved without too much added effort. Now if this was a journal submission this would be easy, the paper would get a revision, and would appear soon, in a better state. Now, with conference submissions, this is a bit more tricky, and I feel quite conflicted here. On the one hand, I quite appreciate the paper building a bridge between deliberation and social choice, and think that this is definitely something that would be appreciated by both communities. On the other hand, it is hard not to look at the paper, and see the potential for improvement, both in terms of strength of the results, but also presentation of the results and I believe this paper could easily be a significantly stronger NeurIPS submission next month.
> >
> > **Our focus on JR**
> > Thank you for checking this. Looks good!
> >
> > **Theoretical results**
> > Thank you for disagreeing, I like this motivation :)
> >
> > ** Set selection and ranking**
> >
> > I still see a slight discrepancy here, especially as there are several works on fair or proportional ranking, but I see what you mean.
> >
> > I personally think that the scores are somewhat meaningless, I will still update it to a 3, and let's see where the reviewer discussion will lead. Thanks again for the nice response. I hope that if the paper gets accepted, the authors take the comments provided by the reviews here seriously (I have been "hurt" far too many times, by authors not doing it...)

---

### Official Review · Reviewer_SQBX · 2025-03-13

**Overall Recommendation:** 4

**Summary:**

The authors propose a comment ranking approach for public deliberation that incorporates justified representation, a concept from the social choice literature. The goal is to rank high quality comments, without losing the representation of groups that are present in the discussion. The approach relies on user approval mechanisms, making it more feasible than algorithms that require external user information. The authors can show that their implementation of enforcing justified representation, using the greedycc algorithm improves representation at a low cost.

**Claims And Evidence:**

The claims made are supported by clear evidence. The experiments resemble a real-world application. There could be some potential bias, because the process relies on user approval (likes or upvotes), which could easily be manipulated in political discussions. This potential drawback is mentioned by the authors themselves and should be investigated in the future.

**Essential References Not Discussed:**

I did not identify essential references that are missing.

**Experimental Designs Or Analyses:**

The experimental designs and analyses are sound. More experiments are needed to investigate potential effects in the future, but since the paper only outlines the approach and theoretical assumptions, this is left for future work.

**Methods And Evaluation Criteria:**

Yes, the proposed methods and evaluation criteria make sense.

**Other Comments Or Suggestions:**

Small errors:
- line 142 (right column): "for all item i" should be "all items i"
- line 203 (right column): "It seems that there could be perverse sets that satisfy JR." I am not sure if the wording is right here this seems off
- line 225 (right column): "the price of JR need to be bounded" should be "needs to be bounded"
- line 226: "need not be compatible" should rather be "does not need to be" or "does not have to be.."

**Other Strengths And Weaknesses:**

The paper is well written and easy to follow. The experiments and theoretical assumptions are clear. The results concerning the use of a score based on the Perspective API are surprising and could also be investigated in the future. It remains unclear if people would acutally feel represented by the algorithms choice and if the algorithm is able to guarantee content-wise diversity. This could be investigated in the future.

**Questions For Authors:**

- Can you give some details about fc (the Perspective API scoring function)? It's an average of all the bridging attribute scores. What is the range of the individual bridging scores?

**Relation To Broader Scientific Literature:**

The authors contribute to the literature on algorithmic fairness by proposing an approach to fair representation that is more feasible in practice than, for example, approaches based on demographics. It is shown that by approximation, sets of comments that satisfy JR can be found at low cost on real-world discussions.

**Theoretical Claims:**

I didn't identify any issues in the proofs in the Appendix.

---

> ### Author Rebuttal · Authors · 2025-03-31
>
> We appreciate the reviewer’s careful reading of our paper and thank them for their constructive comments. We will correct all the typos that they identified.
>
> With regards to the Perspective API bridging score $f_C$, it is the average of the scores for the seven available bridging attributes:
>
> * “Affinity”: References shared interests, motivations or outlooks between the comment author and another individual, group or entity
> * “Compassion”: Identifies with or shows concern, empathy, or support for the feelings/emotions of others.
> * “Curiosity”: Attempts to clarify or ask follow-up questions to better understand another person or idea.
> * “Nuance”: Incorporates multiple points of view in an attempt to provide a full picture or contribute   useful detail and/or context.
> * “Personal Story”: Includes a personal experience or story as a source of support for the statements made   in the comment.
> * “Reasoning”: Makes specific or well-reasoned points to provide a fuller understanding of the   topic without disrespect or provocation.
> * “Respect”: Shows deference or appreciation to others, or acknowledges the validity of another person.
>
> The score for each individual attribute is a probability between 0 and 1. We provide results broken down by each individual attribute in Appendix E.7.
>
> Additional references for the Perspective API bridging classifiers:
> * https://developers.perspectiveapi.com/s/about-the-api-attributes-and-languages?language=en_US
> * https://medium.com/jigsaw/announcing-experimental-bridging-attributes-in-perspective-api-578a9d59ac37
> * Saltz et al. Re-Ranking News Comments by Constructiveness and Curiosity Significantly Increases Perceived Respect, Trustworthiness, and Interest. arXiv 2024.
> * Schmer-Galunder et al. Annotator in the Loop: A Case Study of In-Depth Rater Engagement to Create a Prosocial Benchmark Dataset. AIES 2025.
>
> We thank the reviewer once again for their consideration of our paper.

---

> > ### Comment · Reviewer_SQBX · 2025-04-07
> >
> > Thank you very much for the further clarification. Since I was already convinced that this paper should be accepted, I will not revise my score.

---

### Official Review · Reviewer_Axyg · 2025-03-13

**Overall Recommendation:** 4

**Summary:**

The authors take the problem of content ranking in online social deliberation, and adds justified representation (JR) constraints to the quality optimization problem to ensure diversity and representation. Theoretically, they show that under assumptions of user clusterization, the extra constraint leads to low cost in terms of the quality metric (e.g. civility, engagement). Empirically, they show that adding a greedy implementation of JR significantly improves the representation by existing ranking methods, while imposing only a modest cost in quality.

**Claims And Evidence:**

All claims are backed up by evidence. There are very few logical steps that I find unconvincing (which I detail in later sections).

**Essential References Not Discussed:**

N/A

**Experimental Designs Or Analyses:**

- **[important]** GreedyCC is, in some sense, specifically optimized for the coverage metric. It first searches for a tiny core of comments that satisfy minimum coverage (at least one comment per group), and then fill in the rest with no consideration for representation. Does it remain representative when we use other representation metrics that focuses on not just minimum coverage, but also proportionality? If not, is there another JR approximation algorithm that does well under proportionality?
- It would be helpful to visualize where these methods lie (with & without GreedyCC) on the quality-representation tradeoff plane.
- Have you tried other simple heuristics other than JR, and how well do they work?

**Methods And Evaluation Criteria:**

The authors aim to address an import problem (online public deliberation) with a very interesting approach (content ranking), and give impressive theoretical & empirical evidence to support their approach. I find the approach reasonable and practical.

**Other Comments Or Suggestions:**

- Typo: The y axis of Fig 2 & E.2 are not percentages despite the "%" sign in the axis labels.
- Potentially important as a future direction: The “one member approves one item” condition in JR is too weak (as the authors have acknowledged in the paper). How can we improve this, either theoretically or empirically?

**Other Strengths And Weaknesses:**

- The work is not ground-breaking in the sense that it draws from previous literature on JR and its approximation algorithms. But it gives significant contribution by applying the idea in a important domain, and give a range of theoretical & empirical evidence to back up such an application.

**Questions For Authors:**

I have listed my questions in previous sections. The ones most likely to change my mind are marked with "[important]".

**Relation To Broader Scientific Literature:**

Content ranking, including content ranking with pro-social constraints, is a problem widely studied outside the algorithmic mechanism design literature - for example by RecSys researchers. This paper shows that JR constraint and its variants may be a valuable addition to their toolkit.

**Theoretical Claims:**

- My main uncertainty on the theoretical results lie in the assumptions and how they can be tested on empirical data. Specifically:
  - For theorem 5.1, what is the value $\gamma$ on each of the questions in your dataset?
  - For theorem 5.2, what are the $\gamma$ and $\phi$ values that (statistically speaking) best explains your dataset? This can be done with, for example, a maximum likelihood estimation.
- In reality, there are often cases where each user only reads a very small portion of entries (e.g. someone only reading tweets from people you follow; or someone only upvoting one or two comments before leaving) - do the theoretical bounds remain practical in that case? Would (n/k)-sized cohesive groups exist in that case?
- I did not check the proofs.

---

> ### Author Rebuttal · Authors · 2025-03-31
>
> Thank you to the reviewer for their comprehensive review. We address their questions below.
>
> **Connection Between Theoretical and Empirical Results**
>
> The reviewer asks about how our theoretical results connect to the empirical findings. Even without considering Theorem 5.4, the bound from Theorem 5.1 already provides a close upper bound to the observed prices in practice. We will highlight this connection in our revised text. For Theorem 5.1, the bound holds if there exists an $n/k$-justifying set of size less than or equal to gamma (which the partition of gamma cohesive groups implies). In all ten questions, the GreedyCC algorithm finds an $n/k$-justifying set of size less than or equal to two (see Table E.3). Our bound from Theorem 5.1 would then imply that the price for all three score functions is at most 8/(8-2) = 1.33. The average empirical prices that we observe (see “Enforcing JR comes at a low price” in Section 6) are 1.05 for engagement, 1.06 for diverse approval, and 1.18 for Perspective API.
>
> **Handling Partially Observed Votes**
>
> The reviewer also asks how the theoretical results hold when each user only reads a very small portion of items. In practice, the full approval matrix will need to be inferred from the partially observed votes—which is what we do in our Remesh experiments (see Footnote 10). Since this is a standard recommender system task, we do not focus on this approval inference in our paper. However, when implementing this on a real platform, it is very important to ensure that the inferred approvals are faithful to users’ actual approval. We touch on this issue in our discussion section: “on many platforms, users’ approval will need to be inferred from some form of engagement such as upvotes or likes. If the chosen engagement significantly diverges from actual user approval, the validity of the process could be compromised.”
>
> **Experimental Results and Stronger Proportionality Axioms**
>
> With regards to experimental results, the reviewer asks how our experiments with GreedyCC may hold up against stronger proportionality axioms. In response, we have checked how many of the JR feeds in our experiments also satisfy EJR+ (Brill and Peters, 2023), a much stronger extension of JR. We found that 47 out of 48 feeds also satisfy EJR+ (and will add this result to the paper).
>
> I. Diverse Approval
> - 5/10 DA feeds are JR, all of them are also EJR+
> - 10/10 JR feeds are also EJR+
>
> II. Engagement
> - 3/10 engagement feeds are JR, all of them are also EJR+
> - 9/10 JR feeds are also EJR+
>
> III. Perspective API
> - 10/10 Perspective API feeds are JR, all of them are also EJR+
> - 10/10 JR feeds are also EJR+
>
> This suggests that, although GreedyCC is only optimizing for coverage, it nevertheless satisfies stronger axioms in practice. This result has also been corroborated in other empirical research which has found that JR and EJR+ tend to empirically coincide despite their large theoretical gap (e.g. “Approval-Based Committee Voting in Practice: A Case Study of (Over-)Representation in the Polkadot Blockchain” by Boehmer et al (2024)).
>
> > Potentially important as a future direction: The “one member approves one item” condition in JR is too weak (as the authors have acknowledged in the paper). How can we improve this, either theoretically or empirically?
>
> Empirically, the results just described suggest that our approach may satisfy axioms stronger than JR in real-world datasets. We concur with the reviewer that theoretically studying and guaranteeing stronger extensions of JR would be an interesting direction for future research.
>
> **Other Comments**
>
> > “Have you tried other simple heuristics other than JR, and how well do they work?”
>
> We have not tried other heuristics that are weaker than JR, since the GreedyCC algorithm on its own is already quite simple, and as noted above, it has the advantage of satisfying stronger axioms empirically as well.
>
> > “It would be helpful to visualize where these methods lie (with & without GreedyCC) on the quality-representation tradeoff plane.”
>
> This is a great idea, and we will add these figures to the appendix. Right now, the trade-offs can be understood through the representation results + the prices of JR shown in Figure 2, but we agree that an explicit visualization would make this clearer.
>
> We thank the reviewer again for their thoughtful comments.

---

### Official Review · Reviewer_WMX6 · 2025-03-14

**Overall Recommendation:** 4

**Summary:**

This work applies the principle of "justified representation" as a means to algorithmically surface public comment for the end-goal of public deliberation. This is, in part, motivated by the ideals of deliberative democracy and normative reasons for selected public comments to satisfy some notion of "representativeness." Mathematically, the work studies formalizations of justified representation (and the price of justified representation) from prior work and shows that, assuming that preferences can be modeled as mixtures of Mallows noise model, stronger bounds on the price of JR can be achieved. Finally, the authors run a previously-proposed (approximation) algorithm on real data and show that enforcing JR, even approximately, tangibly improves representation for a variety of score functions $f$.

## Update after rebuttal
My initial review was overall positive; my main concern was about the roles that section 5/6 played in the contribution of the work - the rebuttal provided narrative clarification and I would be happy with seeing this paper in the conference with that discussion incorporated.

**Claims And Evidence:**

* Generally yes, see below for further comments

**Essential References Not Discussed:**

N/A to my knowledge

**Experimental Designs Or Analyses:**

* See above. Main comment is that section 6 is _not_ a direct extension of section 5 and it would be nice to make that clearer; however the experiments themselves are reasonable for what they are.

**Methods And Evaluation Criteria:**

* Typically one would think that the experiments help to evaluate/validate the theory, but that doesn't appear to be the case here, and it's not actually a priori obvious that sections 5 and 6 are rather different and mostly unrelated contributions. That is, section 6 is not validating the theory given in section 5; instead, it is providing experimental evidence with real-world data for the theory work from the prior Elkind work (which does not have experiments). Section 5 is a distinct conceptual contribution about conditions under which we might expect JR to still do well in a utilitarian sense (and could, e.g., even be an explanatory mechanism for what is observed in section 6, though the paper itself doesn't make this connection as far as I can tell). This is not necessarily a problem with the paper itself but perhaps more so with presentation (e.g. if someone was not familiar with Elkind 22 and didn't know that they didn't run real-data experiments, then they might be confused what the message of section 6 is).
* The actual data being used in experiments is highly relevant to the work.

**Other Comments Or Suggestions:**

N/A

**Other Strengths And Weaknesses:**

The paper is overall well-written and easy to follow.

**Questions For Authors:**

* Can you clarify that my interpretation of the roles of sections 5/6 are correct? If so, could you comment on whether you were hoping to achieve something specific by structuring the paper this way?

**Relation To Broader Scientific Literature:**

* This work contributes to concretely realizing the ideal of achieving better (democratic) deliberation, doing so by developing ideas from social choice.
* More specifically, the sufficiency conditions implied by the Mallows mixture analysis are new for this problem setting/ and for JR (as far as I can tell), and build on prior social choice work that studies Mallows mixtures.
* The experiments are followups to prior theory work on JR, and show that previously-proposed algorithms can be practically useful.

**Theoretical Claims:**

* I did not check proofs in detail.
* Based on the statement of Theorem 5.4, I think it could be narratively useful to highlight that the bound is (a) in the worst case over all instances, and (b) therefore not about any particular algorithm or algorithmic strategy; in fact is is unclear how or whether to design algorithms that would have been specific to the Mallows mixture idea.
* A lower bound would be nice (especially wrt identifiability, whether $\gamma$ needs to be known, etc) but I am happy to defer that to future work.

---

> ### Author Rebuttal · Authors · 2025-03-31
>
> We appreciate the reviewer’s thoughtful comments and expertise. The reviewer’s main question is about their interpretation of Sections 5 and 6, which we address in the following. In Section 5, we first introduce novel theoretical results for the price of JR for arbitrary score functions, and then evaluate these results in section 6. Please note that the aspect of arbitrary score functions is key here for the theoretical results and connects both sections.
>
> Our main contribution is to introduce JR into the problem of facilitating deliberation online, particularly on social media platforms. In the social media context, JR is unlikely to be adopted if it also comes at a substantial decrease in user engagement or a substantial increase in the toxicity of comments. Therefore, we analyzed the price of JR, not only with respect to engagement (like Elkind et al did), but also with respect to arbitrary score functions.
>
> Our theoretical results in Section 5 hold for arbitrary score functions, which is why it logically follows to present them before our experiments where we evaluate the price of JR not just for engagement but also for diverse approval and the content-based bridging classifiers. In this way, our experiments also go beyond Elkind et al who only studied engagement. Although the worst-case price for arbitrary score functions can be arbitrarily high (as we showed in Sec 4), the observed price for all three score functions is remarkably low. As the reviewer points out, our theoretical results in Section 5 may explain these findings—a connection we intended to highlight and will clarify in our revision.
>
> Our empirical results also contribute to the literature on prosocial interventions in recommender systems, such as bridging-based ranking. The most common way to operationalize “bridging” is via diverse approval. However, surprisingly, we found that the content-based bridging classifiers provided much better representation than diverse approval. Indeed, all the Perspective API bridging feeds always satisfied JR by default. This is surprising as there is a line of literature showing that *toxicity* classifiers (including Perspective API’s) can be biased against various groups (e.g. Sap et al 2019, Lee et al 2024). Yet, it appears that the newer *bridging* classifiers target attributes with broad appeal, suggesting that these content-based classifiers may be a promising direction for future research in prosocial ranking.
>
> We will also make clear that the bound in Theorem 5.4 is not about any particular algorithmic strategy.
>
> Thank you again to the reviewer for your careful review of our paper.

---

> > ### Comment · Reviewer_WMX6 · 2025-04-03
> >
> > Thanks, this is useful clarification and I hope some discussion to this end can be added to the final version of the paper!

---

### Decision · Program_Chairs · 2025-05-01

**Decision:**

Accept (poster)

**Comment:**

This paper addresses the problem of fair representation in comment ranking by introducing a constraint, justified representation (JR), drawn from social choice theory. The authors show, both theoretically and empirically, that incorporating this constraint improves representation without compromising other metrics such as conversational quality or user engagement.

This work lies at the intersection of deliberation theory, social choice, and algorithmic ranking.

Following the rebuttal phase, which helped clarify remaining misunderstandings, all reviewers agreed that the paper's claims are supported by clear and convincing evidence. The methods and evaluation are solid, and the experiments are sound and valid. While the paper is well-positioned within the literature, one reviewer noted that the contribution might be somewhat limited from a purely social choice perspective. Nonetheless, the work offers a valuable bridge between deliberation and social choice.

The paper could be further improved by clarifying the interpretation of Sections 5 and 6, and by including a visualization of the methods with & without GreedyCC on the quality-representation tradeoff plane.

Overall, I believe this is a valuable contribution to the conference and recommend acceptance.

The reviewer discussion phase was particularly helpful to make explicit suggestions for improvement, which I strongly encourage the authors to incorporate into the revised version. These suggestions focus on making the limitations of the proposed methodology more explicit:

- The JR axiom does not fully imply the "idealized" notion of proportionality described in the footnote. However, as the authors note, this seems to have little impact in their experiments.

- The paper would benefit from a more thorough contextualization with respect to the broader social choice literature, particularly regarding alternative fairness notions and the rationale for not adopting them in this work.

- The authors should be more transparent about the limitations of their results. For example, acknowledging that the idea of mixing rules to achieve approximate guarantees was already present in earlier work on this topic.